**TECHNIQUES AND RESOURCES**

SPECIAL ISSUE
LIFELONG DEVELOPMENT

# A single-cell transcriptomic atlas of sensory-dependent gene expression in developing mouse visual cortex

Andre M. Xavier[1,*], Qianyu Lin[1,*], Chris J. Kang[1] and Lucas Cheadle[1,2,‡]

## ABSTRACT

Sensory experience drives the maturation of neural circuits during postnatal brain development through molecular mechanisms that remain to be fully elucidated. One likely mechanism involves the sensory-dependent expression of genes that encode direct mediators of circuit remodeling within developing cells. To identify potential drivers of sensory-dependent synaptic development, we generated a single-nucleus RNA sequencing dataset describing the transcriptional responses of cells in the mouse visual cortex to sensory deprivation or to stimulation during a developmental window when visual input is necessary for circuit refinement. We sequenced 118,529 nuclei across 16 neuronal and non-neuronal cell types isolated from control, sensory deprived and sensory stimulated mice, identifying 1268 sensory-induced genes within the developing brain. While experience elicited transcriptomic changes in all cell types, excitatory neurons in layer 2/3 exhibited the most robust changes, and the sensory-induced genes in these cells are poised to strengthen synapse-to-nucleus crosstalk and to promote cell type-specific axon guidance pathways. Altogether, we expect this dataset to significantly broaden our understanding of the molecular mechanisms through which sensory experience shapes neural circuit wiring in the developing brain.

KEY WORDS: Sensory experience, Visual cortex, Circuit development, Transcription, Neuron, Synapse, Cell signaling

## INTRODUCTION

The precise connectivity of neural circuits in the mammalian brain arises from a convergence of genetic and environmental factors. Brain circuits are first assembled *in utero* via the formation of an overabundance of synaptic connections between neurons, then later remodeled, or refined, postnatally through the strengthening of some of these synapses and the elimination of others (Katz and Shatz, 1996; Hong and Chen, 2011). The selective retention of a subset of initially formed synapses equips the brain with an interconnected network of circuits optimized to facilitate neurological function across the lifespan. Furthermore, impairments in synaptic refinement are increasingly appreciated

to contribute to a host of neurodevelopmental conditions, such as autism and schizophrenia (Zoghbi and Bear, 2012; Neniskyte and Gross, 2017; Feinberg, 1982). Thus, elucidating the mechanisms underlying circuit refinement in the early postnatal brain is important from both basic and translational perspectives.

While much emphasis has been placed on defining the intrinsic genetic mechanisms that govern embryonic stages of brain development, less is known about how environmental cues shape the maturation of neural circuits postnatally. A prime example of environmentally driven circuit development can be seen in the role of sensory experience in refining neural circuitry within the mouse visual system around the third week of life (Hooks and Chen, 2020). Specifically, between postnatal days (P)20 and P30, visual experience promotes the structural and functional refinement and maturation of synaptic connections between excitatory thalamocortical neurons within the dorsal lateral geniculate nucleus (dLGN) of the thalamus and their postsynaptic targets in layer 4 of primary visual cortex (V1) (McGee et al., 2005; Mataga et al., 2004; Coleman et al., 2010; Zhou et al., 2017). Importantly, blocking visual experience during this developmental window by rearing mice in complete darkness significantly impedes the maturation visual circuits, whereas blockade of experience outside of this time frame does not have a strong effect (Hooks and Chen, 2008, 2006; Sato and Stryker, 2008). Thus, sensory experience drives the development of thalamocortical circuits during a defined window of time in V1.

While the visual system has provided numerous insights into functional aspects of sensory-driven synaptic refinement, our understanding of the molecular mechanisms that mediate this process remains limited. One likely mechanism is the induction of gene programs in neurons in response to sensory-driven neuronal activity, a process termed activity-dependent transcription (Yap and Greenberg, 2018). In this process, synaptic innervation drives the influx of $Ca^{2+}$ into the postsynaptic cell through ionotropic glutamate receptors and L-type $Ca^{2+}$ channels (Bading et al., 1993), leading to the phosphorylation of transcription factors in the nucleus, such as CREB and MEF2 (Kornhauser et al., 2002; Flavell et al., 2008). Within the first hour of synaptic innervation, these newly activated factors induce the expression of immediate-early genes (IEGs), many of which encode a separate set of transcription factors, including the well-established IEG *Fos* (Malik et al., 2014). During a second wave of activity-dependent transcription, which typically occurs several hours following neuronal activation, IEGs bind a subset of genomic elements to drive the expression of a second cohort of genes (late-response genes, LRGs) encoding direct mediators of synaptic remodeling, such as the secreted neurotrophin *Bdnf* (Kim et al., 2010; Hong et al., 2008). This two-wave pattern of activity-dependent transcription encompassing the early expression of IEGs followed by the later expression of LRGs is likely to contribute to the refinement of visual circuits at the molecular level.

[1]Cold Spring Harbor Laboratory, Cold Spring Harbor, NY 11724, USA. [2]Howard Hughes Medical Institute, Cold Spring Harbor, NY 11724, USA.
*These authors contributed equally to this work

‡Author for correspondence (cheadle@cshl.edu)

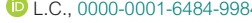 L.C., 0000-0001-6484-9984

Moreover, previous work indicates that sensory-driven gene programs in the brain are highly cell type-specific, reflecting the capacity of activity-dependent genes to shape cellular function in a precise manner (Hrvatin et al., 2018; Cheadle et al., 2018; Mardinly et al., 2016). Thus, genes that are induced by neurons in response to sensory experience are promising candidates to mediate sensory-driven circuit development in V1.

Given the potential of sensory-induced gene programs to represent a key molecular link between visual experience and circuit development, we generated a single-cell transcriptomic atlas of sensory-driven gene expression in mouse V1 during the period of vision-dependent refinement taking place the third week of life. We assessed inducible gene programs across 118,529 individual nuclei representing 16 major cortical cell types through differentially expressed gene (DEG) analysis and RNA velocity analysis, and we derived insights into molecular interactions between cells using the computational tool *CellChat*. We expect this dataset to be a valuable resource for investigators interested in uncovering the molecular basis of sensory-dependent synapse remodeling and plasticity in the developing brain.

## RESULTS

### A visual deprivation and stimulation paradigm for capturing sensory-induced transcripts

To characterize the gene programs that are elicited by sensory experience during postnatal brain development, we harnessed a dark-rearing method to manipulate experience in mice in a temporally restricted manner. In this paradigm, mice were initially reared according to a standard 12-h light/12-h dark cycle (normal rearing, NR) before being placed in complete darkness at P20, which is when sensory-dependent visual circuit refinement begins. Mice were then maintained in a completely dark environment for 24 h a day until P27, when sensory-dependent refinement peaks. At this time, one cohort of mice was sacrificed in the dark without re-exposure to light, and V1 tissue was collected. Other cohorts of dark-reared mice were acutely re-exposed to light at P27 for varying lengths of time, a manipulation that leads to the acute activation of circuit refinement and plasticity in both the dLGN and V1 (Hooks and Chen, 2020, 2008; Thompson et al., 2016). Thus, this late dark-rearing (LDR) paradigm allowed us to assess the impact of (1) sensory deprivation and (2) sensory stimulation on gene expression in developing V1 (Fig. 1A).

Previous studies of sensory-dependent transcription in the adult visual system have analyzed IEG induction 1 h following re-exposure of dark-reared animals to light (Hrvatin et al., 2018). To confirm that 1 h was also optimal for capturing IEG induction during development, we performed qPCR and single-molecule fluorescence *in situ* hybridization (smFISH) in parallel on V1 tissue after subjecting mice to the LDR paradigm, assessing the expression of the canonical IEGs *Fos* and *Jun* as a read-out. We found that the expression of both IEGs was increased as early as 15 min after light re-exposure, and that this increase persisted for at least 2 h following stimulation. Within this time frame, the peak of *Fos* and *Jun* expression occurred not at 1 h but at 30 min after light re-exposure (Fig. 1B-D). Thus, we included a 30 min stimulation timepoint to capture IEGs in our experiments, alongside three additional timepoints at which we expected to capture the bulk of LRGs: 2 h, 4 h and 6 h. Altogether, our finalized dataset includes cells from mice according to the following six conditions: normally reared (NR) mice at P27; mice reared in complete darkness from P20 to P27 (LDR); and mice reared in complete darkness between P20 and P27 then acutely re-exposed to light for 30 min (LDR30m), 2 h (LDR2h), 4 h (LDR4h) or 6 h (LDR6h).

### Mapping sensory-dependent gene expression in the developing cortex

To map sensory-dependent changes in gene expression across cortical cell types in an unbiased manner, we performed single-nucleus RNA sequencing (snRNAseq; 10X Genomics) on V1 tissue bilaterally micro-dissected from mice following the LDR paradigm described above (Fig. 1A). We sequenced individual nuclei rather than cells based upon our interest in capturing nascent transcriptional events that are acutely induced by experience. Three biological replicates were performed for each condition, with each replicate including cells pooled from the visual cortices of three animals to increase yield. Biological replicates were collected, isolated and processed independently on different days to control for batch effects. After next-generation sequencing, the data were mapped to the mouse genome and quality control was performed to remove putative doublets and unhealthy or dying cells from the dataset using Seurat and DoubletFinder packages in R (McGinnis et al., 2019; Hao et al., 2021). Data were then integrated across biological replicates and conditions for downstream analysis within Seurat. The final dataset includes 118,529 nuclei across 16 distinct cell clusters, representing eight excitatory neuron subtypes, four inhibitory neuron subtypes and four glial subtypes (Fig. 1E-G). The excitatory populations captured include layer 4 (L4) excitatory neurons, layer 2/3 (L2/3) excitatory neurons, three populations of layer 5 (L5) neurons and three populations of layer 6 (L6) neurons. Inhibitory populations sequenced include Grin3a-enriched neurons (some of which also express SST markers), parvalbumin (PV) neurons, VIP neurons and neurons expressing NPY, which include neurogliaform cells and a subset of SST neurons. Glial populations sequenced include astrocytes, oligodendrocytes, oligodendrocyte precursor cells (OPCs) and microglia (Fig. 1H,I). Cell type assignments were based upon the presence of marker genes identified previously (Hrvatin et al., 2018; Yao et al., 2021). The numbers of cells within each cell class included in the dataset are given in Table S1.

### Sensory deprivation upregulates a cohort of genes in excitatory neurons

With this dataset in hand, we set out to understand how manipulating sensory experience impacts the transcriptional states of cells in V1 during development. To this end, we used the DEseq2 function within Seurat to identify transcripts that were significantly differentially expressed (differentially expressed genes, DEGs; false discovery rate (FDR)<0.05) between each condition for each cell type, beginning with a comparison of gene expression in normally reared (NR) mice at P27 versus sensory deprived (i.e. LDR) mice at the same age. This analysis revealed changes in gene expression meeting a minimum threshold of $\log_2(1.5)$ fold change in excitatory neurons following dark-rearing compared to NR mice. Specifically, when the DEG analysis was applied to all excitatory neuron clusters in aggregate, 52 genes were found to be downregulated in the NR condition compared to LDR, indicating that depriving mice of light increased the expression of a defined cohort of genes (Fig. S1A). Among the excitatory neuron clusters, the subtypes that exhibited the largest numbers of gene expression changes following LDR were L2/3 neurons (86 genes upregulated in LDR and 7 genes upregulated in NR; Fig. S1B) followed by neurons in the L6a cluster (21 genes upregulated in LDR; Fig. S1C). In L2/3 neurons, genes more highly expressed in the LDR condition included factors such as *Tspan11* and *Gpc3*, which are involved in cellular dynamics and migration (Huang et al., 2022; Akkermans et al., 2022). Interestingly, the seven genes that were upregulated in the NR condition included known activity-

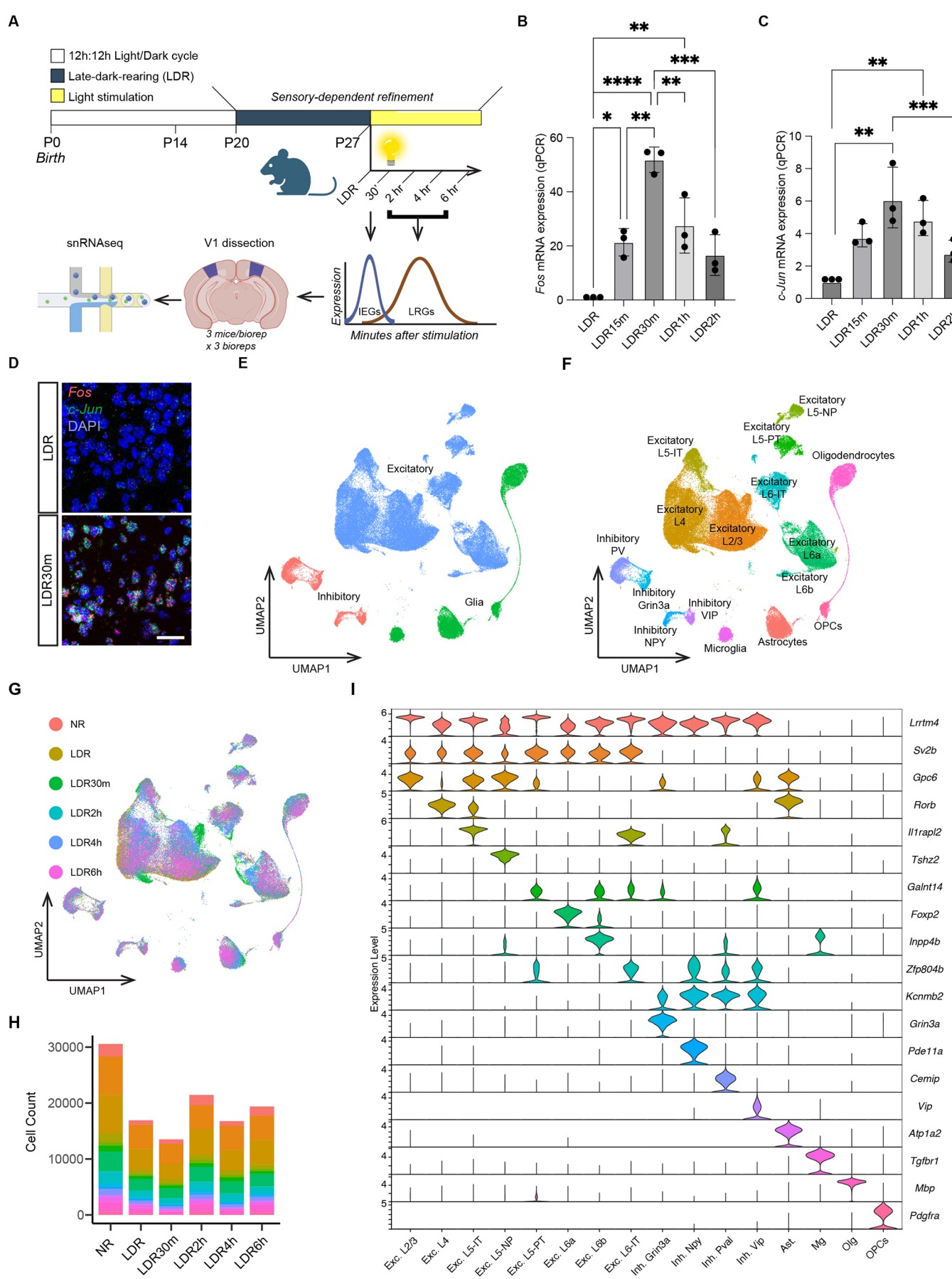

**Fig. 1.** See next page for legend.

**Fig. 1. Experimental design and introduction to the single-nucleus RNA sequencing dataset.** (A) Schematic illustrating the late dark-rearing (LDR) paradigm and the workflow of the single-nucleus RNA sequencing (snRNAseq) experiments. (B) Quantification of *Fos* mRNA expression in sensory deprived (LDR) mice and in mice acutely exposed to light for between 15 min and 2 h, with stimulation timepoints labeled as follows: LDR15m (15 min of light), LDR30m (30 min), LDR1h (1 h) and LDR2h (2 h). *Fos* expression assessed by qPCR and normalized to *Gapdh*. Values plotted are additionally normalized to the LDR condition. (C) qPCR quantification of *Jun* mRNA expression (normalized to *Gapdh*) in V1 across all timepoints. Data obtained by qPCR and values plotted are normalized to LDR. In B and C, data are mean±s.e.m. *n*=3 mice per condition. One-way ANOVA followed by Tukey's post hoc test: *P<0.05; **P<0.01; ***P<0.001; ****P<0.0001. (D) Example confocal images of V1 in sections from a sensory deprived mouse (LDR) and a mouse re-exposed to light for 30 min (LDR30m). *Fos* mRNA (red), *Jun* mRNA (green) and DAPI (blue). Scale bar: 44 µm. (E) UMAP plot illustrating the 118,529 nuclei in the dataset categorized by general cell class: excitatory neurons (blue), inhibitory neurons (pink) and glia (green). (F) UMAP plot with all 16 clusters colored and labeled by cell type. (G) UMAP plot with cells colored by condition according to the legend on the left. (H) Numbers of cells of each type included in the final dataset across all conditions. See also Table S1. (I) Violin plot demonstrating the enrichment of markers used to assign nuclei in the dataset to distinct cell types. The top enriched gene per cluster is listed on the *y*-axis on the right; normalized FPKM expression given on the *y*-axis on the left; cluster identity shown on the *x*-axis.

regulated factors such as the neurotrophin *Bdnf* and the nuclear orphan receptor *Nr4a1*. In contrast to excitatory neurons, only one gene, the synaptically localized long non-coding RNA *Gm45323*, was downregulated in the NR compared to the LDR condition (Niu et al., 2023) (Fig. S1D). These findings suggest that excitatory neurons are more sensitive to the effects of sensory deprivation than inhibitory neurons at the transcriptional level, and indicate that sensory deprivation tends to increase, rather than decrease, gene expression in these cells. Overall, however, these transcriptomic changes were relatively subtle; as a result, differences in gene expression between cells from animals re-exposed to light at any given stimulus timepoint versus cells from either the LDR or NR condition are largely overlapping (Tables S2 and S3).

### Excitatory and inhibitory neurons mount shared and distinct transcriptional responses to sensory stimulation

We next compared DEGs between each light re-exposure timepoint (LDR30m, LDR2h, LDR4h and LDR6h) and the sensory-deprived LDR condition for all 16 clusters in the dataset. These experiments revealed bidirectional changes in gene expression at every timepoint analyzed within most cell types, yielding a total number of 1268 genes that are upregulated at any stimulation timepoint compared to LDR (Fig. S2A,B and Table S2). These genes included numerous known IEGs, such as the AP1 factors *Fos* and *Jun* (which were broadly expressed across numerous cell types and cortical layers), the neuron-specific IEG *Npas4*, and the Nr4a and Egr families of TFs that are induced by various extracellular stimuli, including synaptic innervation (Yap and Greenberg, 2018) (Fig. 2A-E). Interestingly, although AP1 transcription factors are broadly considered to be IEGs, *Jund* and *Junb* (which are distinct from *Jun*) exhibited a pattern of induction more consistent with an LRG identity, peaking at LDR2h rather than LDR30m (Fig. 2A). Among all cell types analyzed, sensory experience elicited the most robust gene expression changes in L2/3 excitatory neurons, followed by excitatory neurons in L6 (L6a and L6-IT) and L4 (Fig. S2A). This pattern persisted even when cell populations were first downsampled to contain the same number of cells prior to DEG analysis (Fig. S3). That L2/3 neurons exhibit the largest number of transcriptional

changes as a result of light re-exposure is consistent with a recent report identifying L2/3 cells as being particularly sensitive to sensory experience during postnatal development (Cheng et al., 2022).

Given the heightened responsiveness of L2/3 neurons to sensory experience, we next isolated the 27,482 L2/3 neurons in the dataset and performed an additional round of clustering on this population alone. We identified six sub-populations of L2/3 neurons, including four clusters that were made up of cells from at least two timepoints (clusters 0, 1, 2 and 3) and two clusters that were predominantly made up of cells from a single timepoint (clusters 4 and 5) (Fig. S4A,B). Specifically, cluster 4 was primarily composed of cells from the LDR30m condition (78%) while cluster 5 was primarily composed of cells from LDR6h (97%; Fig. S4C,D). To ensure that these results were consistent across replicates, we measured the numbers of cells from each replicate found within clusters 4 and 5. While cluster 4 contained a significant number of cells from all three biological replicates (877, 821 and 536 cells per replicate 1-3, respectively), cluster 5 almost exclusively contained cells from replicate 1. Thus, we conclude that visual stimulation can give rise to a distinct transcriptomic state of L2/3 cells specifically at 30 min after light stimulation.

In addition to IEGs, we also identified cohorts of genes that were preferentially upregulated at LDR2h, LDR4h or LDR6h, which fits the expected profile of LRGs (Fig. 2F). We found that the vast majority of excitatory neurons, Grin3a+ and VIP+ inhibitory neurons, and astrocytes underwent the most transcriptional changes at LDR30m, while NPY+ inhibitory neurons, PV+ inhibitory neurons, microglia and oligodendrocytes induced the most genes at LDR6h (Fig. S2A). While some of these gene programs overlapped across cell types, many were cell-type-specific, suggesting that sensory-induced genes can orchestrate unique functions within distinct cell types.

To further probe the dataset, we next aggregated cells across all excitatory neuron classes or inhibitory neuron classes, and assessed how these broad populations responded to experience at the transcriptomic level. We found that light re-exposure elicited the most robust changes in gene expression for excitatory neurons at LDR30m (233 genes upregulated), while the most robust changes in inhibitory neurons (82 genes upregulated) occurred 6 h after light re-exposure (Fig. 2G). This observation could reflect a temporal trajectory in which excitatory neurons are more strongly impacted by sensory stimulation first, with inhibitory neurons responding later.

We next determined the overlap between the DEGs that were upregulated by stimulation at each time point across aggregated inhibitory and excitatory clusters. Unexpectedly, of the 233 genes that are upregulated in excitatory neurons following light re-exposure at LDR30m, only 31 (13.3%) were also upregulated in inhibitory cells at the same timepoint. Conversely, 47 (66%) of the 71 genes upregulated in excitatory neurons at LDR6h were shared with inhibitory neurons (Fig. 2H,I). These findings suggest the possibility that the LRG programs within these cell types may have more in common than earlier waves of sensory-induced transcription, which could be a pattern that is specific to developing V1.

Gene ontology (GO) analysis was then applied to compare the functional classifications of sensory-driven transcription for inhibitory and excitatory cells, revealing similarities in the types of genes induced by each cell type. For example, at LDR30m, DEGs in both classes were enriched for GO categories such as 'RNA polymerase II-specific DNA-binding transcription factor binding', reflecting the sensory-induced expression of members of the Nr4a family in both excitatory and inhibitory cells. Conversely, GO

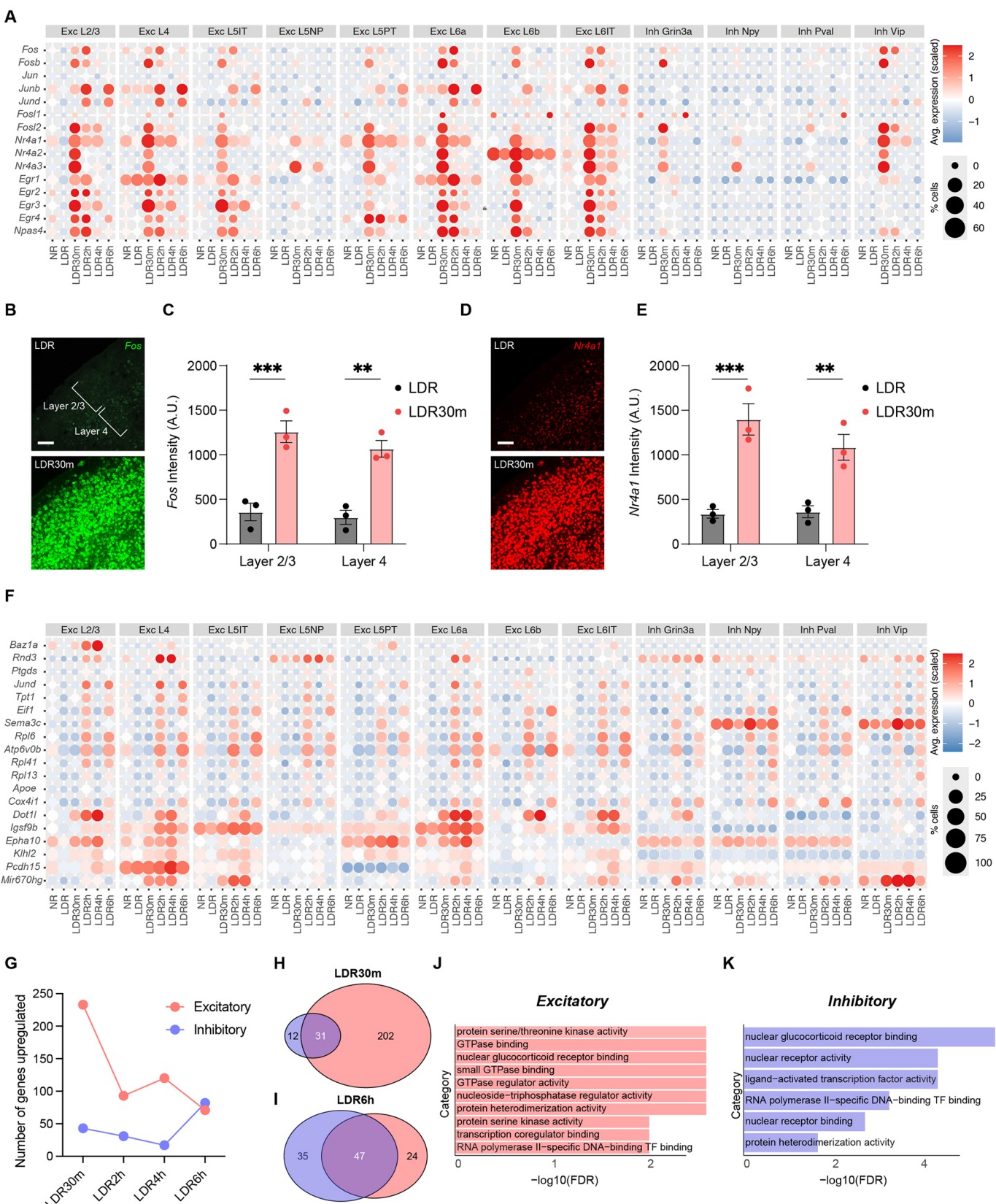

**Fig. 2.** See next page for legend.

categories related to 'GTPase binding' and 'GTPase regulator activity', including the Rho GTPase guanine nucleotide exchange factors (RhoGEFs) *Arhgef3* and *Plekhg5*, were selectively upregulated in excitatory neurons at this timepoint (Fig. 2J,K). Overall, these data indicate that the gene programs induced in excitatory and inhibitory neurons downstream of sensory

**Fig. 2. Excitatory and inhibitory neurons mount shared and distinct responses to sensory stimulation.** (A) Bubble plot illustrating the induction of canonical immediate-early genes (IEGs) across timepoints and cell types. Color indicates relative expression level according to the scale on the right. Size of circle represents the percentage of cells expressing the gene. (B) Confocal images of V1 in late dark-reared (LDR) mice and in mice re-exposed to light for 30 min (LDR30m) subjected to single molecule fluorescence *in situ* hybridization (smFISH) to label *Fos* mRNA. Scale bar: 100 µm. (C) Quantification of *Fos* expression (arbitrary units, A.U.) in L2/3 and L4 of V1 in LDR and LDR30m mice. Data are mean±s.e.m. Two-way ANOVA with Tukey's post-hoc test: \*\*$P<0.01$, \*\*\*$P<0.001$; $n=3$ mice/ condition. (D) Confocal images of V1 in LDR and LDR30m mice subjected to smFISH to label *Nr4a1* mRNA. Scale bar: 100 µm. (E) Quantification of *Nr4a1* expression in L2/3 and L4 in LDR and LDR30m mice. Data are mean ±s.e.m. Two-way ANOVA with Tukey's post-hoc test: \*\*$P<0.01$, \*\*\*$P<0.001$; $n=3$ mice/condition. (F) Bubble plot demonstrating late-response gene (LRG) expression across cell types and conditions. Scaled expression is indicated on the right. (G) Graph displaying the numbers of genes significantly upregulated at each stimulation timepoint (compared to LDR control) across conditions for aggregated excitatory (pink) and inhibitory (purple) neurons. (H,I) Venn diagrams demonstrating overlap between sensory-dependent gene programs in excitatory (pink) versus inhibitory (purple) neurons at LDR30m (H) and LDR6h (I). (J,K) Gene ontology (GO) categories enriched among genes upregulated in excitatory (J) and inhibitory (K) neurons at LDR30m.

stimulation exhibit partial overlap at each time point analyzed, with the amount and nature of overlap varying significantly by condition.

### Comparison of sensory-induced genes in L2/3 and L4 excitatory neurons reveals a shared protein kinase signature and divergent axon guidance pathways

Given that L2/3 and L4 neurons were among the most strongly impacted by sensory experience, we next compared sensory-driven gene programs between these populations. For both cell types, we found that LDR30m was the peak of DEG expression (303 genes upregulated in L2/3 neurons and 239 genes in L4 neurons) followed by LDR4h, when 210 and 124 genes were upregulated in L2/3 and L4 neurons, respectively (Fig. 3A-F). We next assessed the overlap between the gene programs induced by L2/3 and L4 neurons at each timepoint. Of the combined unique genes upregulated at LDR30m across both cell classes, 193 (or 55%) were induced in both cell types (Fig. 3G). Varying degrees of overlap were also observed between sensory-dependent gene programs at the later timepoints as follows: 31% overlap at LDR2h, 36% at LDR4h and 52% at LDR6h (Fig. 3H-J). Thus, L2/3 and L4 neurons mounted both shared and distinct responses to sensory experience that were most robust at LDR30m followed by LDR4h.

We next investigated the nature of the sensory-dependent gene programs induced by each cell types using GO analysis. As expected, several of the same categories emerged for L2/3 and L4 neurons, including 'GTPase binding' (likely reflecting mechanisms of cytoskeletal remodeling) and 'nuclear receptor binding' (associated with activity-dependent transcription factors), but the 'protein serine/threonine kinase activity' category was particularly prominently represented in both cell types (Fig. 3K-N). Many of the genes associated with this category [e.g. the extracellular signal-regulated (ERK)-family kinases *Mapk4* and *Mapk6*, and the Salt-inducible kinases *Sik1*, *Sik2* and *Sik3*] are known to interact with numerous activity-induced transcription factors identified in the dataset (Jennings et al., 2020; Melgarejo da Rosa et al., 2016; Jiang et al., 2024; Proschel et al., 2017). Thus, genes upregulated by sensory experience in L2/3 and L4 neurons share a protein kinase signature that we predict may strengthen synapse-nucleus crosstalk following sensory stimulation principally in excitatory neurons.

We next performed GO analysis on the gene sets that were uniquely induced in each cell type. An interesting pattern to emerge was the differential induction of two axon guidance pathways within these populations: the ephrin pathway (including ephrin receptors *Ephb3* and *Epha10*) in L2/3 neurons and the semaphorin pathway (including the semaphorin co-receptors *Plxna4* and *Nrp1*) in L4 neurons (Table S2). Both of these pathways mediate the migration of neuronal axons and the establishment of synapses within target zones based upon ephrin and semaphorin ligand expression, and have been implicated in establishing retinotopy in the developing visual system (Sweeney et al., 2015; Triplett and Feldheim, 2012; Claudepierre et al., 2008; Prieur et al., 2023). Utilizing FISH, we validated that, consistent with the snRNAseq analysis, *Ephb3* is upregulated in L2/3 but not L4 in LDR30m mice, and we demonstrated that a major ligand of the *Ephb3* receptor, ephrin B3 (*Efnb3*), is most highly expressed within layers 4-6 of V1 (Fig. S5). Given that L2/3 neurons are known to directly project axons and form synapses within deeper layers of V1, this result is consistent with the possibility that L2/3 neurons engage sensory-dependent EphB3/Efnb3 signaling to remodel their connections in response to sensory stimulation, although this hypothesis should be functionally tested in follow-up studies. Altogether, these findings suggest that sensory experience may elicit axonal remodeling and/or presynaptic plasticity by inducing the expression of members of two distinct signaling families, ephrins and semaphorins, in L2/3 and L4 neurons, respectively.

### Sensory-induced transcripts in inhibitory and glial cell types

In addition to identifying sensory-induced changes in excitatory neurons, the dataset also revealed stimulus-dependent gene programs across multiple classes of inhibitory cells. Among the four inhibitory neuron classes represented in the dataset, PV+ neurons and Grin3a+ neurons exhibited the most transcriptional changes following sensory stimulation (101 and 95 genes upregulated at any given timepoint, respectively, compared to LDR), while NPY+ and VIP+ neurons induced only 16 and 47 genes total, respectively (Fig. S2A and Table S2). Interestingly, the temporal trajectory of sensory-dependent transcription varied among interneuron classes such that, while the greatest numbers of upregulated DEGs were observed at LDR30m for Grin3a+ and VIP+ neurons (similar to excitatory neurons; Fig. 4A,B), PV+ and NPY+ interneurons upregulated the most genes at LDR6h, suggesting that some inhibitory neuron types may respond to sensory stimulation more slowly than others (Fig. 4C-F). While several of the same canonical IEGs observed in excitatory neurons were induced in numerous inhibitory classes at LDR30m, such as the Nr4a family of transcription factors and the synaptic scaffold *Homer1*, other genes were upregulated in only a subset of cell types. For example, we employed multiplexed FISH to validate the upregulation of the synaptic modifier brain-derived neurotrophic factor (*Bdnf*) in excitatory L4 neurons but not in VIP+ inhibitory neurons (Fig. 5A,B,K). Similarly, we validated the upregulation of the stress-associated cue corticotropin-releasing hormone (*Crh*) in VIP+ neuron but not in L4 excitatory, PV+ inhibitory or NPY+ inhibitory neurons at LDR30m (Fig. 5C-F,L). Finally, we validated the upregulation of the long non-coding RNA *Dlx6os1* in both VIP+ and NPY+ neurons, but not in L4 excitatory or PV+ inhibitory cells (Fig. 5G-J,M). Thus, inhibitory neurons mount transcriptomic responses to sensory stimulation, with some DEGs overlapping and some DEGs distinct between cell types.

The inclusion of glial cells in our dataset provided the opportunity to explore how sensory experience influences non-neuronal brain cells at the transcriptomic level. We found that,

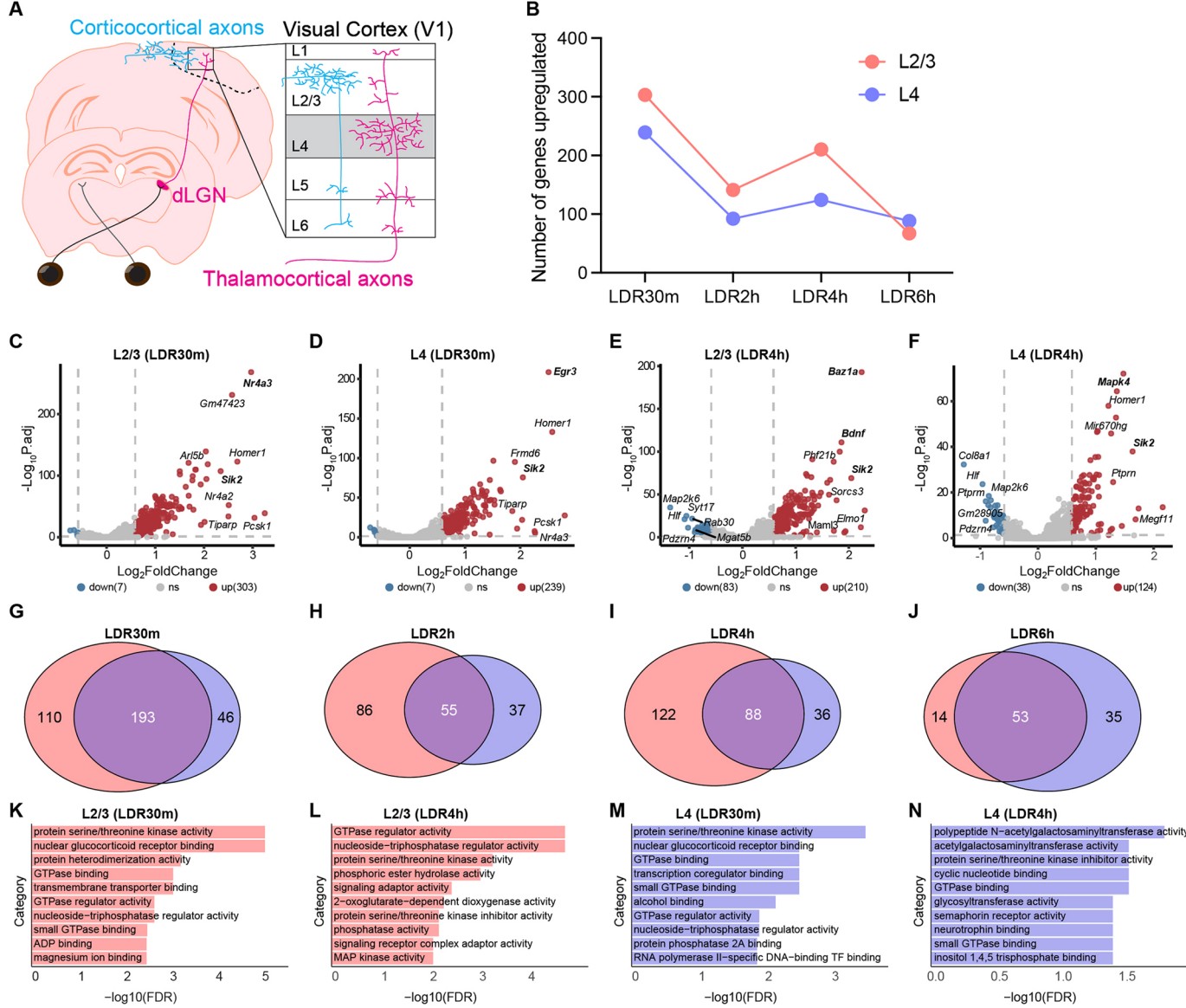

**Fig. 3. Comparison of sensory-driven gene expression in L2/3 and L4 excitatory neurons reveals a shared protein kinase signature and divergent axon guidance pathways.** (A) Schematic of the pathway from the retina to the primary visual cortex (V1) in the mouse. L2/3 neurons principally receive 'top-down' input from other regions of cortex (blue), while L4 neurons receive 'bottom-up' inputs from visual thalamus (magenta). (B) Graph displaying the numbers of genes significantly upregulated at each stimulation timepoint [compared to late dark-reared (LDR) control] across conditions for L2/3 (pink) and L4 (purple) neurons. (C,D) Volcano plots illustrating genes that were significantly upregulated (red) or downregulated (blue) in L2/3 (C) and L4 (D) neurons after 30 min of light re-exposure following LDR. Genes of particular interest are in bold. (E,F) Volcano plots illustrating genes that were significantly upregulated (red) or downregulated (blue) in L2/3 (E) and L4 (F) neurons after 4 h of light re-exposure following LDR. (G-J) Venn diagrams displaying overlap between upregulated genes identified in L2/3 (pink) versus L4 neurons (purple) at the LDR30m (G), LDR2h (H), LDR4h (I) and LDR6h (J) timepoints. (K,L) Gene ontology (GO) analyses of genes upregulated by light in L2/3 neurons at LDR30m (K) and LDR4h (L). (M,N) GO analysis of genes upregulated by light in L4 neurons at LDR30m (M) and LDR4h (N).

among glial cell types, astrocytes mounted the most elaborate responses to light re-exposure, upregulating 82 genes at LDR30m and 49 genes at LDR6h (Fig. 4G,H). This pattern of inducing more genes early on and fewer genes later in the timecourse is consistent with what we observed for the vast majority of excitatory neuron cell types (Fig. S2A). Astrocytes were the only glial cell type to follow this pattern, which could reflect the close coupling of astrocytic activity and synaptic transmission, and may also be related to roles for astrocytes in visual cortical development (Allen and Eroglu, 2017; Singh et al., 2016). In contrast, microglia and oligodendrocytes upregulated more genes at LDR6h (20 genes and 63, respectively) than at LDR30m (1 and 5, respectively),

suggesting that, similar to PV+ and NPY+ inhibitory neurons, these cells respond to sensory stimulation more slowly than the majority of excitatory neurons (Fig. 4I,J). An interesting pattern to emerge was the upregulation of the lipoprotein-associated factor *Apoe* in all three glial cell types at LDR6h, potentially reflecting shared pathways induced by experience across cell types (Fig. 4H-J). Conversely, we observed the upregulation of the complement component *C1qa*, a molecule that mediates the elimination of synapses through phagocytosis, only in microglia at LDR6h (Fig. 4I). This finding could reflect a role for sensory-induced transcription in the activity-dependent elimination of synapses by microglia during a developmental period when experience is known to drive synaptic

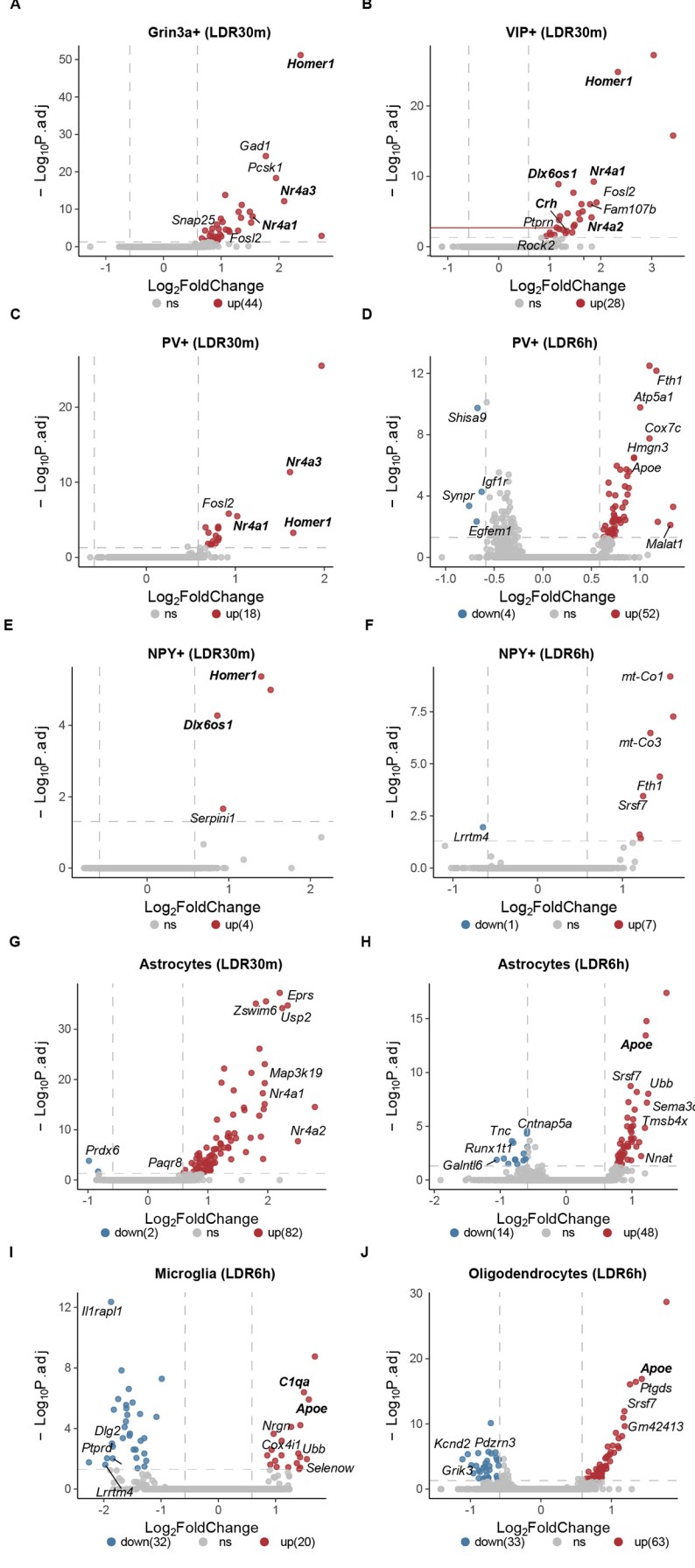

**Fig. 4. Sensory-dependent gene expression in inhibitory neurons and glia.** (A-J) Volcano plots demonstrating transcripts that were significantly differentially expressed (differentially expressed genes, DEGs) in inhibitory neurons (A-F) and glia (G-J) following light re-exposure after dark rearing. Y-axis, negative Log(10) adjusted P value (threshold of P.adj<0.05 indicated by dashed horizontal line); x-axis, Log(2) fold change [threshold of log₂(1.5) indicated by dashed vertical lines]. Red, genes that are upregulated by experience; blue, genes that are downregulated by experience; gray, genes that are unchanged by experience. Genes of particular interest (i.e. mentioned in the text) are in bold.

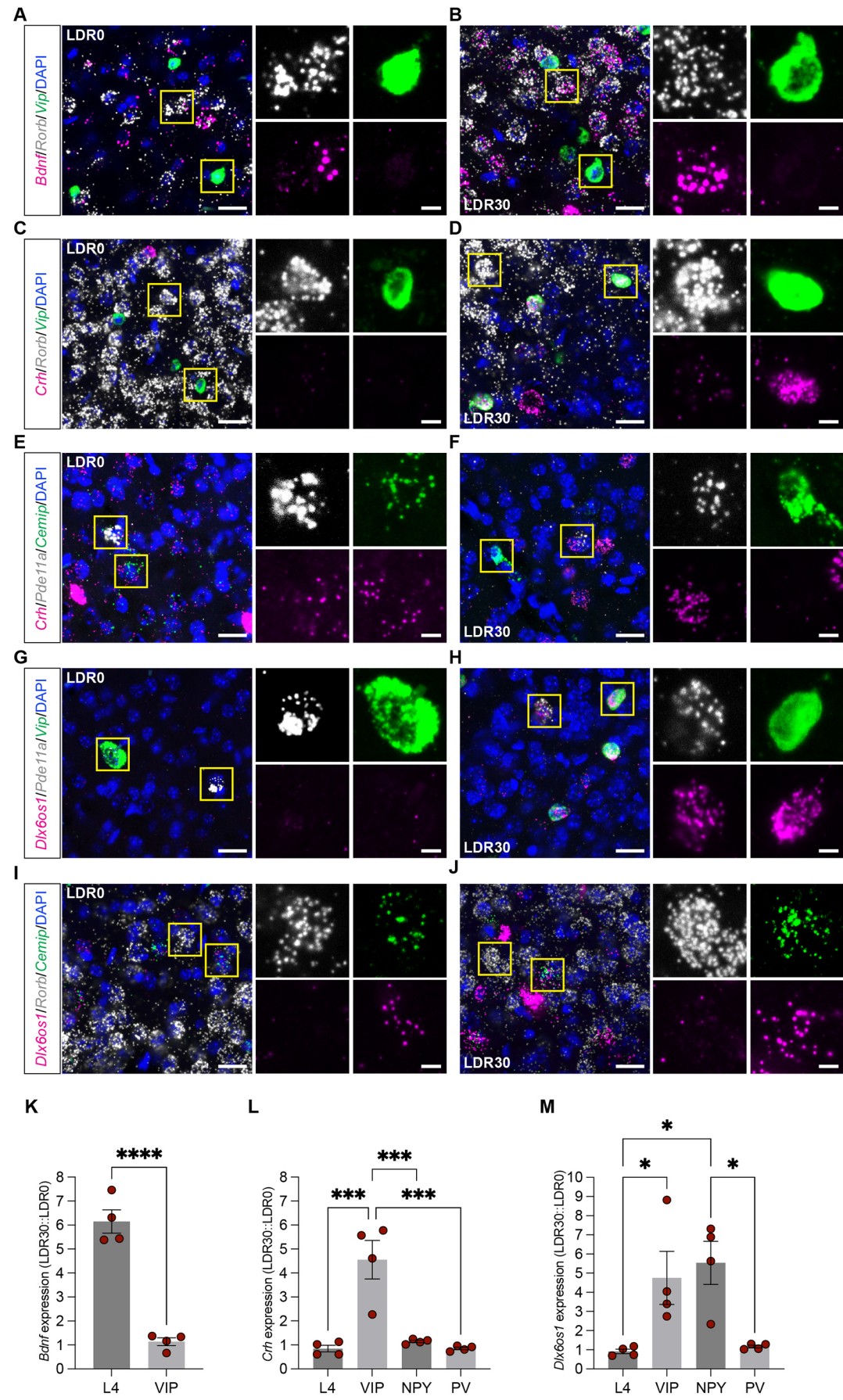

**Fig. 5.** See next page for legend.

**Fig. 5. *In situ* validation of the cell type-specific induction of *Bdnf*, *Crh* and *Dlx6os1*.** (A-J) Example confocal images of coronal sections of visual cortex subjected to fluorescence *in situ* hybridization and probed for *Bdnf*, *Crh* and *Dlx6os1* as follows: *Bdnf* (magenta) expression in *Rorb+* excitatory L4 neurons (white) and *VIP+* inhibitory neurons (green) at LDR0 (A) and LDR30m (B); *Crh* (magenta) expression in L4 neurons (white) and *VIP+* neurons (green) at LDR0 (C) and LDR30m (D); *Crh* expression (magenta) in *Pde11a+* NPY inhibitory neurons (white) and *Cemip+* PV inhibitory neurons (green) at LDR0(E) and LDR30m (F); *Dlx6os1* (magenta) expression in *NPY+* neurons (white) and *VIP+* neurons (green) at LDR0 (G) and LDR30m (H); and *Dlx6os1* (magenta) expression in L4 neurons (white) and PV neurons (green) at LDR0 (I) and LDR30m (J). Scale bars: 20 μm (left); 5 μm (right). (K) Quantification of *Bdnf* expression in L4 versus VIP neurons plotted as fluorescence intensity averaged across at least six cells per biological replicate at LDR30m divided by intensity at LDR0; *n*=4 where each biological replicate is one mouse. Data are mean±s.e.m. Unpaired Student's *t*-test: ****$P<0.0001$. (L,M) Similar quantifications of *Crh* expression (L) and *Dlx6os1* expression (M) in L4, VIP, NPY and PV neurons. Data are mean ±s.e.m. One-way ANOVAs followed by Tukey's post-test: *$P<0.01$, ***$P<0.001$.

refinement. Our dataset is consistent with previous work which suggests that sensory-dependent gene expression is not restricted to excitatory and inhibitory neurons but also occurs in glial cells (Hrvatin et al., 2018; Cheadle et al., 2020).

## Insights into sensory-dependent gene induction and repression dynamics in L2/3 and L4 neurons based upon RNA velocity

Increases in RNA abundance following sensory stimulation are often interpreted to reflect the new transcription of genes. However, RNA abundance can be influenced by many mechanisms beyond transcription, such as changes in the stability or degradation of the mRNA. To identify genes that were most likely to be upregulated following light stimulation via a transcriptional mechanism, we applied RNA velocity analysis using the UniTVelo package. Briefly, this approach estimates transient cell state transcriptional dynamics based upon the relative abundance of nascent (unspliced) and mature (spliced) mRNAs (Svensson and Pachter, 2018; La Manno et al., 2018; Gao et al., 2022). Although this type of analysis is more commonly performed on whole-cell RNA sequencing data, which better captures post-transcriptional regulation of mature mRNAs typically occurring in the cytosol, we found that our data contained a sufficient amount of both mature and immature RNAs to employ the assay as previously reported (Adewale et al., 2024; Marsh and Blelloch, 2020).

For L2/3 and L4 neurons, we analyzed the architecture of transcript maturation for each predicted cell state transition: LDR to LDR30m, LDR30m to LDR2h, LDR2h to LDR4h, and LDR4h to LDR6h. These comparisons revealed strong signatures of both transcriptional induction and repression in L2/3 and L4 neurons with a stereotyped pattern shared by both cell types (Fig. 6A). For example, between LDR and LDR30m, relatively large numbers of genes in each cell type (204 and 161 genes in L2/3 and L4 cells, respectively) exhibited transcriptional induction, with only very few genes exhibiting repression. On the contrary, between LDR30m and LDR2h, the majority of significantly altered genes were repressed. Between LDR2h and LDR4h, most altered genes were induced, although many genes were also repressed. Finally, between LDR4h and LDR6h, the majority of altered genes in each cell type exhibited a repressed profile (Fig. 6B,C). These results are in line with the canonical view of stimulus-dependent gene programs involving primarily two waves of transcription: an IEG wave peaking at LDR30m; and a LRG wave peaking at LDR4h.

To more fully understand the dynamics underlying sensory-dependent transcription, we next asked whether the genes that are induced at LDR30m exhibit sustained expression across the timecourse, or whether their expression returned to normal by LDR6h. Among the 204 genes that were induced in L2/3 neurons between LDR and LDR30m, 112 (55%) were repressed between LDR30m and LDR2h (Fig. 6D). A similar comparison in L4 neurons revealed that 38% of genes induced between LDR and LDR30m are repressed between LDR30m and LDR2h (Fig. 6E). We next compared the dynamics of genes that were upregulated at the LDR4h timepoint, which our data suggest is the peak of LRG programs in both L2/3 and L4 neurons. We observed that, among the 132 genes induced between LDR2h and LDR4h in L2/3 neurons, 98 genes (74%) were repressed between LDR4h and LDR6h (Fig. 6F). The same analysis in L4 neurons revealed that, of the 45 genes that were induced between LDR2h and LDR4h, 30 (67%) were repressed between LDR4h and LDR6h (Fig. 6G). These data suggest that a significant proportion of the genes that were induced at LDR30m or LDR4h were repressed within 2 h of induction. These data highlight distinct cohorts of genes in L2/3 and L4 neurons that exhibit transcriptional induction and/or repression dynamics within the time window captured in our paradigm (Table S4). Supporting the consistency of our results, we found that about 33% of the genes that were upregulated between LDR and LDR30m in L2/3 and L4 neurons based upon DEG analysis were also induced between these time points when assessed by RNA velocity (Fig. 6H,I). Hence, the RNA velocity analysis uncovered subsets of DEGs that we speculate are most likely to represent bona fide IEGs and LRGs, based upon their transcriptional dynamics.

## Inference of cell-cell interactions using CellChat uncovers neurexin and neuregulin signaling in developing V1

Cells in the brain interact dynamically with one another not only through contact-mediated mechanisms but also through molecular signaling between compatible ligand-receptor pairs. However, a systematic catalog of intercellular interactions in developing visual cortex was lacking. Thus, we next harnessed our snRNAseq dataset to analyze putative cell-cell interactions in V1 across all cell types using the computational tool CellChat, which harnesses databases of known ligand-receptor binding partners to estimate the number and strength of putative intercellular communication pathways based upon gene expression data (Jin et al., 2021). Applying CellChat to the NR condition within the snRNAseq dataset, we detected 442 significant ligand-receptor pairs among the 16 cell clusters captured (Fig. 7A,B). We further categorized these pairs as belonging to 58 discrete signaling pathways. Consistent with sensory experience promoting synaptic maturation during the time window analyzed, modules related to synapse development and plasticity were among the strongest pathways identified. For example, the neurexin (Nrxn) family of autism-linked presynaptic adhesion molecules that mediate synapse function by binding neuroligins (Nlgns) and leucine-rich repeat transmembrane neuronal proteins (Lrrtms) at postsynaptic specializations was the strongest signaling pathway uncovered by CellChat. Signaling between neuregulins (Nrgs) and ErbB receptors, which orchestrates the formation of excitatory synapses onto inhibitory neurons (Muller et al., 2018; Ting et al., 2011), was the second most enriched module identified. Apart from Nrxn and Nrg signaling, Ncam and Cadm (i.e. SynCAM1) adhesion molecules were also identified as active signals in V1. Furthermore, consistent with axonal remodeling occurring during sensory-dependent refinement, EphA and EphB ephrin receptors and semaphorins 3-6 were also predicted to signal actively (Fig. 7C,D). These findings

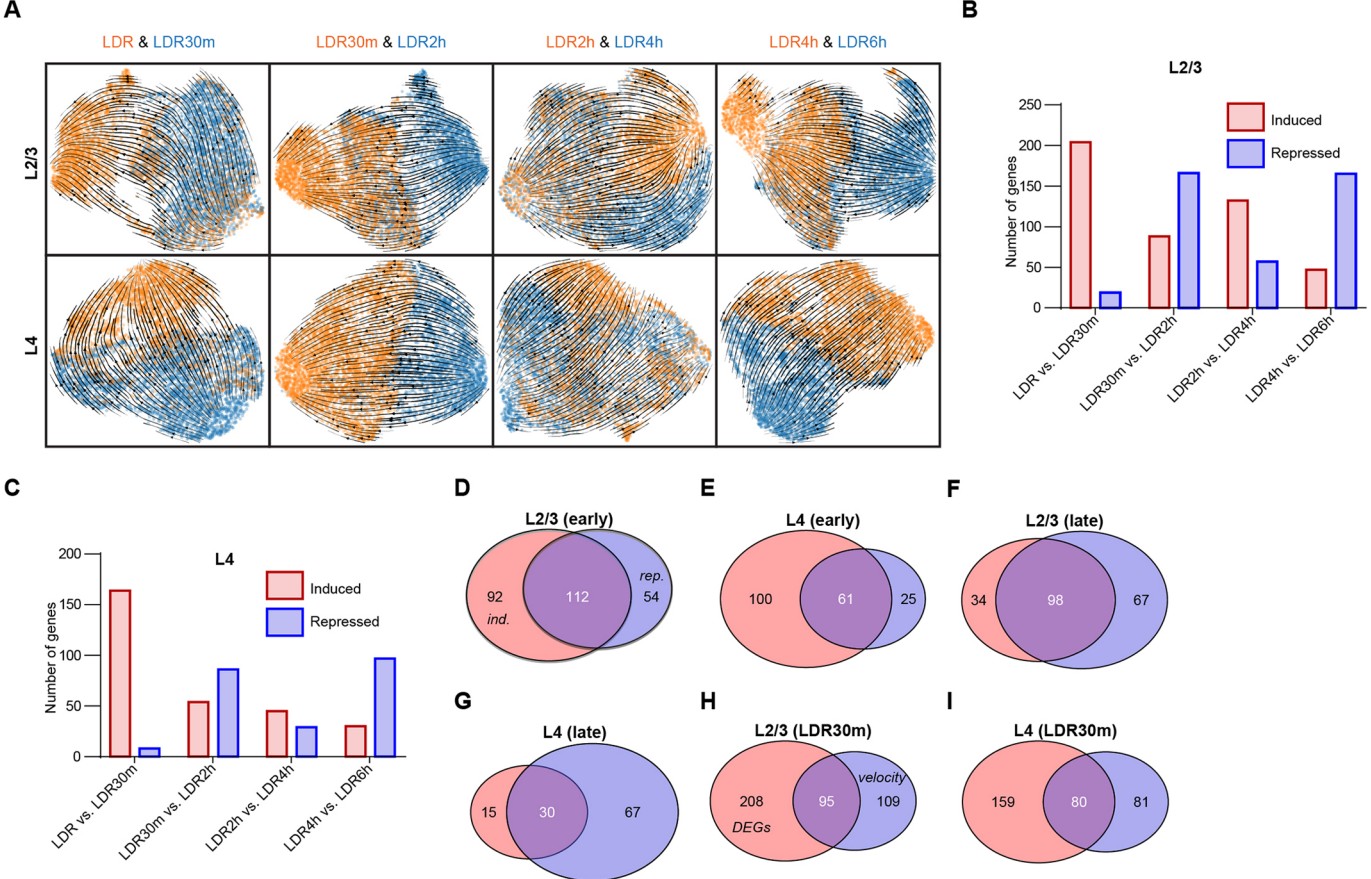

**Fig. 6. Transcriptional induction and repression events in L2/3 and L4 neurons revealed by RNA velocity.** (A) UMAP plots generated based upon RNA velocity displaying transcriptional dynamics across each cell-state transition. L2/3 neurons, top row, L4 neurons, bottom row. (B,C) Bar graphs displaying the total numbers of induced (red) and repressed (blue) genes across each cell-state transition in L2/3 (B) and L4 (C) neurons. (D,E) Venn diagrams displaying overlap between the genes induced at LDR30m (red) and the genes that are repressed between LDR30m and LDR2h (blue) in L2/3 (D) and L4 (E) neurons. (F,G) Venn diagrams displaying overlap between the genes induced between LDR2h and LDR4h (red) and the genes that are repressed between LDR4h and LDR6h (blue) in L2/3 (F) and L4 (G) neurons. (H,I) Overlap between upregulated DEGs and induced genes in L2/3 (H) and L4 (I) neurons at LDR30m.

suggest that cells in V1 work together to shape developing circuits in response to sensory experience via molecular signaling pathways that converge upon synapses.

We next assessed the putative contributions of the different cell types in V1 to the Nrxn and Nrg signaling pathways identified via CellChat. The primary outgoing signals of the Nrxn pathway were Nrxns 3 and 1, and they were most prominently expressed by L6b excitatory neurons (Fig. 7C and Fig. S6A). The primary receivers of these signals were Nlgn1 and Lrrtm4, which were most highly expressed in L5-PT neurons but also appeared in L4 neurons and, to a lesser extent, in other populations as well (Fig. 7D and Fig. S6A). In general, we found that excitatory neurons were more heavily involved in both the propagation of outgoing and the receipt of incoming molecular signals than inhibitory neurons or glia, with neurons in L6 being particularly active (Fig. S6A,B). This result is in line with excitatory neurons in L6 being among the cell types that exhibited the most transcriptional changes following sensory stimulation (Fig. S2). Interestingly, while the inducible gene programs in L2/3 and L4 excitatory neurons shared many features (Fig. 3), these cell classes differed substantially in their predicted participation in cell:cell signaling, with L4 neurons being much more likely to participate in signaling with other V1 cells than L2/3 neurons. Among inhibitory populations, NPY-expressing cells were the strongest senders of outgoing signals, while VIP neurons

were the strongest receivers (Fig. S6A,B). In contrast, several excitatory populations were predicted to produce Nrg, with L6b neurons being the most prominent expressers followed by L4 neurons. All inhibitory cells were predicted to be relatively strong receivers of Nrg signaling, except for VIP+ neurons (Fig. S6B; Table S5). Overall, these data highlight the utility of the snRNAseq resource described here to uncover important principles underlying the molecular control of circuit maturation in the developing brain.

## DISCUSSION

Since the seminal work of Nobel laureates David Hubel and Torsten Wiesel in the 1960s (Wiesel and Hubel, 1965, 1963), sensory experience has been known to be a major driver of brain development. However, our understanding of the molecular mechanisms engaged by experience to shape brain wiring has remained limited. While molecular adaptations at individual synapses, such as changes in neurotransmitter receptor composition, are well poised to mediate the effects of activity within an acute time frame, in a developmental context, more global adaptations are warranted. Accordingly, the idea that robust changes in gene expression driven by sensory stimulation during brain development may play a vital role in circuit refinement is consistent with emerging evidence that neurons in the visual cortex undergo significant epigenetic and genomic changes across the first

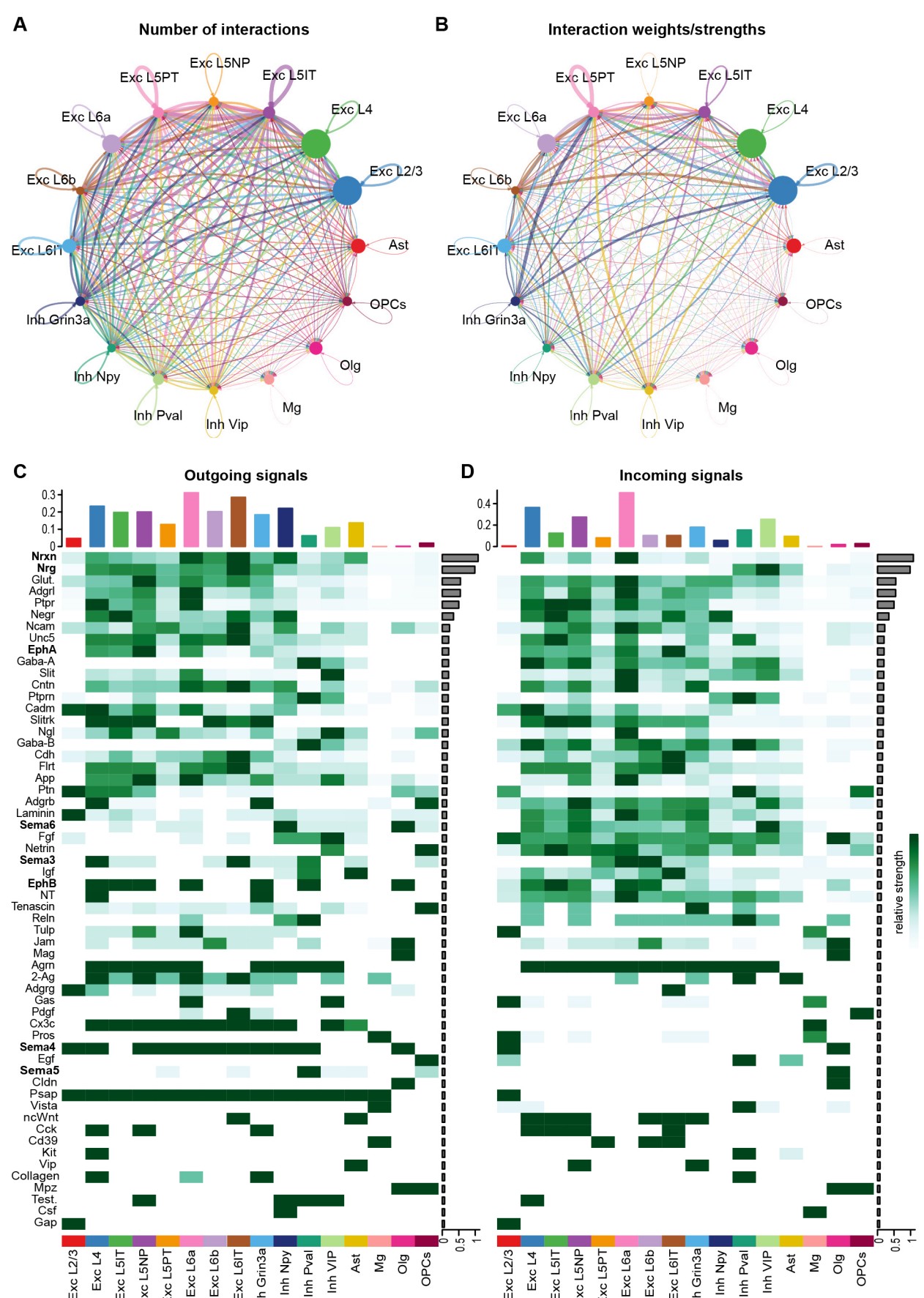

**Fig. 7.** See next page for legend.

**Fig. 7. Inference of putative cell:cell interactions in developing V1 using CellChat.** (A) Cellular communication plot demonstrating the predicted numbers of intercellular ligand-receptor interactions between all cell types in the dataset. (B) Comparative weights/strengths of the predicted cell:cell interactions plotted in A. (C,D) Heatmaps displaying distinct cell signaling modules (*y*-axis, pathways of interest in bold) predicted by CellChat across all cell types (*x*-axis) in the dataset. Top: bars representing the contributions of each cell type to outgoing (C) or incoming (D) signals aggregated across signaling modules. Bar graphs on the right of each heatmap demonstrate the contribution of each individual signaling pathway to the overall interaction score generated in CellChat. Heatmap colors indicate the relative strength of signaling activity of a given pathway, as predicted by CellChat, according to the scale on the right.

month of life in mice, including between P20 and P27 (Stroud et al., 2020, 2017). Because these changes in gene expression occur at the cellular rather than the synaptic level, they are likely to exert substantial influence over the development and maintenance of circuits in the long term. Thus, we expect the single-cell RNA sequencing dataset presented here to reveal crucial insights into the factors that underlie the maturation of neural circuits in the postnatal brain.

Several observations that we have made in interrogating this dataset may be of particular interest for future studies. For example, the observation that L2/3 and L4 neurons strongly upregulate intracellular signaling molecules such as protein serine/threonine kinases as early as 30 min after stimulation suggests that sensory-dependent gene programs in these cells may reinforce synapse-to-nucleus crosstalk. Furthermore, the finding that L2/3 neurons (but not L4 neurons) induce *Ephb3* expression, coupled with the finding that the ligand of *Ephb3*, *Efnb3*, is preferentially expressed in deep layers of V1 that are known to receive projections from neurons in L2/3, may suggest that experience promotes axonal remodeling and plasticity in a cell type-specific manner. In addition, the observation that excitatory neurons are likely more sensitive to experience than inhibitory cells, both at the level of sensory-induced gene expression changes as well as cell signaling interactions, could increase our understanding of the differential roles that these cell types play in visual function. Among excitatory neurons, the discovery that L2/3 neurons are particularly strongly affected is consistent with a recent study highlighting that the maturation of these cells is influenced by visual experience (Cheng et al., 2022). At the level of cell signaling, our data showing that the strongest signatures were related to Nrxn and Nrg signaling pathways suggests that cellular interactions within developing V1 converge upon synapses.

While the RNA Velocity analysis is informative, it is important to consider caveats in interpreting these data. Our analysis relies upon single-nucleus RNA sequencing, which predominantly captures immature pre-mRNAs (for the present dataset, we captured ~82% immature RNAs and ~18% mature RNAs) and limits the ability to detect post-transcriptional regulation occurring on mature mRNAs, as much of this occurs in the cytoplasm. Additionally, the library preparation method we used enriched for polyadenylated RNA, reducing the representation of nascent RNAs that are crucial for robust RNA velocity modeling. As a result, the insights provided by RNA velocity analysis in this study are valuable in their consistency with the results of the DEG analysis, but strong conclusions about transcription-independent regulatory mechanisms should not be drawn from these experiments. These limitations highlight the complementary nature of RNA velocity analysis to other approaches, rather than serving as a standalone method for studying regulatory mechanisms.

In the future, computational comparisons between the results of our study and the results of transcriptomic studies of inducible gene expression in adult mice could shed light on development-specific mechanisms of sensory-driven circuit remodeling. For example, Hrvatin et al. (2018) previously published a single-cell dataset of sensory-driven gene expression in adult V1 utilizing the same LDR paradigm as that used here, providing a potential opportunity to integrate their dataset with ours to rigorously assess the differences and their relevance to developmental biology. That said, the Hrvatin study used whole-cell rather than single-nucleus sequencing, and they assessed immediate-early gene expression at 1 h rather than 30 min. Thus, future comparisons between these two datasets, as long as these caveats are taken into account, could provide valuable insights into development-specific aspects of sensory-dependent transcription.

## MATERIALS AND METHODS
### Animals
All experiments were performed in compliance with protocols approved by the Institutional Animal Care and Use Committee at Cold Spring Harbor Laboratory (CSHL). Male C57Bl/6J mice were obtained from the Jackson Laboratory (000664) then housed at CSHL in an animal facility where average temperatures and humidity were maintained between 20 and 21°C, and 54-58%, respectively. Mice in this study were aged between P18 and P27. Animals had access to food and water *ad libitum*.

### Late dark-rearing paradigm
Male C57Bl/6J mice were obtained from the Jackson laboratory at P18 and allowed to acclimate to the standard 12-h light/12-h dark environment of the CSHL animal facility until P20, at which point they were separated into six cohorts. One cohort was maintained under normal housing conditions (normally reared, NR) while the other cohorts were placed inside a well-ventilated,100% light-proof chamber (Actimetrics). Mice in the chamber were housed in complete darkness until P27, at which point one cohort was euthanized and perfused with ice-cold 1×PBS (snRNAseq experiments) or 1×PBS followed by 4% PFA (smFISH experiments) in the dark. The remaining four cohorts of mice were also dark reared between P20 and P27, but were then re-exposed to light for varying lengths of time: 30 min [late dark rearing (LDR)30m], 2 h (LDR2h), 4 h (LDR4h) and 6 h (LDR6h). After perfusing the mice and removing their brains in the dark, V1 regions were micro-dissected from all cohorts in the wet lab.

### Single-nucleus RNA sequencing and data analysis
#### V1 tissue collection
Whole brains were placed into ice-cold 1×Hank's balanced salt solution (HBSS) supplemented with Mg$^{2+}$ and Ca$^{2+}$. The V1 brain regions were then bilaterally micro-dissected under a 3.5×-90× Stereo Zoom microscope (AmScope) using a needle blade. Micro-dissected tissue was either immediately processed for snRNAseq or was frozen for later processing. For each experimental condition, V1 region tissues from three mice were pooled prior to nuclear suspension preparation, library preparation and sequencing. The use of *n*=3 mice per condition was selected to improve statistical power for DEG analysis, following standard practices for transcriptomics studies. Additionally, in our experience with snRNAseq and scRNAseq, *n*=3 replicates is sufficient to uncover robust and reproducible features of biology.

#### Nuclear suspension preparation
The V1 tissue was transferred to a 1 ml dounce homogenizer containing 300 µl of ice-cold supplemented homogenization buffer (0.25 M Sucrose, 25 mM KCl, 5 mM MgCl$_2$, 20 mM Tricine-KOH, 5 mM DTT, 0.75 mM Spermine, 2.5 mM Spermidine, 0.05× Protease Inhibitor Cocktail, 1 U/µl of RNAse Inhibitor and 0.15% IGEPAL CA-630). Note the inclusion of drugs to block gene transcription and protease activity, as well as a RNase inhibitor to protect the integrity of the RNA. The tissue was homogenized with a loose pestle then a tight pestle about 10-15 times each. The samples were then filtered using a 20 µm filter.

## Library construction and sequencing

Single-cell gene expression libraries were prepared using the Single Cell 3′ Gene Expression kit v3.1 (10× Genomics, 1000268) according to manufacturer's instructions. Libraries were sequenced on an Illumina Nextseq2000 to a mean depth of ~30,000 reads per cell.

## Raw data processing

The raw FASTQ files were processed using Cell Ranger (v7.1.0) and aligned to the mm10 reference mouse genome. Loom files for cell dynamics analysis were generated using Velocyto (v0.17.17) by mapping BAM files to the gene annotation GTF file (refdata-gex-mm10-2020-A). Each library derived from the single-nucleus datasets underwent identical processing, resulting in a gene expression matrix of mRNA counts across genes and individual nuclei. Each cell was annotated with the sample name for subsequent batch correction and meta-analysis.

## Quality control, cell clustering and cell type annotation

To ensure the integrity of our single-cell RNA sequencing data, we implemented several quality control measures. First, we calculated the log10 of the number of genes per UMI (log10GenesPerUMI), and cells with a value less than 0.85 were excluded. We also removed cells with more than 1% mitochondrial gene expression to reduce noise from apoptotic or damaged cells. Additional thresholds included excluding cells with fewer than 500 UMIs or 300 genes to eliminate low-quality or empty droplets. Doublets were identified and excluded using the DoubletFinder package, with optimal pK values determined for each sample through a sweep analysis (McGinnis et al., 2019). Following these steps, we applied the standard Seurat (v4) pipeline for data pre-processing (https://satijalab.org/seurat/articles/get_started.html), which included selecting the top 3000 highly variable genes and regressing out UMI counts and mitochondrial gene percentage for cell clustering.

Clustering utilized the functions FindNeighbors and FindClusters from Seurat, employing resolutions ranging from 0.1 to 0.5. A resolution of 0.5 was ultimately selected for clustering. To identify major cell types, the ConserveredMarkers function (log2 fold change>0.25, MAST test, adjusted $P$-value<0.05 with Bonferroni correction), with pct.1>70% and pct.2<30% identified unique and highly enriched differentially expressed genes (DEGs) in specific clusters compared to others. Cell types were manually annotated based on the expression of conserved markers (Hrvatin et al., 2018; Yao et al., 2021), ensuring precise identification and accurate analysis of cellular phenotypes.

## Differentially expressed gene analysis

Differentially expressed genes (DEGs) between conditions were identified using the DEseq2 function within Seurat v4. To perform DEG analysis, we first generated pseudo-bulk data by aggregating expression counts for each cell type within each sample using the AggregateExpression function in Seurat. DEGs were then identified for each of the 16 individual clusters included in the dataset.

## RNA velocity analysis

Cell velocity analysis was conducted on L2/3 and L4 excitatory neurons using the UniTVelo (v0.2.4) tool within the scVelo (v0.2.5.) framework, focusing on the 2000 most variably expressed genes. Genes were categorized based on their fit_t scores, such that those with a fit_t>0 were classified as induced genes, whereas genes with a fit_t <0 were identified as repressed genes.

## Cell-cell interaction analysis

The R package CellChat (http://www.cellchat.org/) was used to infer cell-cell interactions within our dataset. We adhered to the standard pipeline and default parameters set by CellChat. The complete CellChatDB.mouse database was employed, which categorizes ligand-receptor pairs into 'Secreted Signaling', 'ECM-Receptor' and 'Cell-Cell Contact'. Additionally, we conducted CellChat analyses on the overall dataset and separately for conditions at specific timepoints – LDR0, LDR30m, LDR2h, LDR4h, LDR6h and NR – although we focus on the NR condition in this paper.

## Enrichment analysis

Gene Ontology (GO) enrichment analysis was conducted using the 'clusterProfiler' (v4.10.0) package. For the analysis of differentially expressed genes (DEGs), only genes with an adjusted $P$-value less than 0.05 and a log2 fold change greater than log2(1.5) were included. For the analysis of induced and repressed genes, all identified genes were considered. The parameters for the GO analysis were set with a $P$-value cutoff of 0.05 and a q-value cutoff of 0.2, using the Benjamini-Hochberg (BH) method for adjusting $P$-values. This approach ensures rigorous identification of biological processes significantly associated with the gene sets under study.

## Real-time qPCR

Flash-frozen V1 samples were processed for RNA extraction using Trizol (ThermoFisher, 15596018) according to the manufacturer's protocol. The cDNA library was built using iScript Kit (BioRad, 1725037) and Oligo d(T) primers. The real-time PCR were performed using SybrGreen kit (Fisher, A25742) and standard PCR temperature protocol. *Fos* and *Jun* expression were normalized to *Gapdh* levels. The following primer sequences were used: *Fos* (forward), 5′-GGGAATGGTGAAGACCGTGTCA-3′; *Fos* (reverse), 5′-GCAGCCATCTTATTCCGTTCCC-3′; *Jun* (forward), 5′-CAGTCCAGCAATGGGCACATCA-3′; *Jun* (reverse), 5′-GGAAGCG-TGTTCTGGCTATGCA-3′; *Gapdh* (forward), 5′-CATCACTGCCACC-CAGAAGACTG-3′; *Gapdh* (reverse), 5′-ATGCCAGTGAGCTTCCCG-TTCAG-3′.

## Single-molecule fluorescence *in situ* hybridization

Animals were anesthetized with a ketamine and xylazine cocktail (ketamine, 90 mg/kg; xylazine, 10 mg/kg) before perfusion with ice-cold phosphate-buffered saline (PBS) followed by 4% paraformaldehyde (PFA) in 1×PBS. Brains were then drop-fixed in 4% PFA in 1×PBS for 24 h. Brains were then washed with 1×PBS thrice for 10 min before being transferred to a 30% sucrose solution at 4°C. After dehydration, brains were embedded in optimal cutting temperature (OCT; VWR, 25608-930) and stored at −80°C. Coronal sections (20 μm) containing the visual cortex were cut using a cryostat and thaw-mounted onto a Superfrost Plus microscope slide (Thermo Fisher Scientific, 1255015) and stored at −80°C until the experiment. Fluorescence *in situ* hybridization (FISH) was performed using the RNAScope platform V2 kit (Advanced Cell Diagnostics (ACD), 323100) according to the manufacturer's protocol for fixed-frozen sections. Samples were then counterstained with DAPI before ProLong Gold Antifade was applied. A 1.5× thickness coverslip was then applied to the slides, which were then stored at 4°C until imaging. Commercial probes from ACDBio (RNAscope) were obtained to detect the following genes: *Fos* (316921), *Nr4a1* (423342-C2), *Jun* (453561-C3), *Pde11a* (481841), *Vip* (415961-C3), *Meis2* (436371-C3), *Dlx6os1* (1195751-C2), *Crh* (316091-C2), *Bdnf* (424821-C2), *Cemip* (438231-C3), *Ephb3* (510251), *Efnb3* (526771) and *Rorb* (444271 and 444271-C3).

## Confocal imaging

Single-molecule fluorescence in situ hybridization (smFISH) images were acquired using the Zeiss LSM780 with a 20×/0.8 n.a. objective or a 63×/1.4 n.a. objective. Z-stack images were acquired.

## Fluorescence *in situ* hybridization quantification

To assess *Fos* and *Nr4a1* expression, FISH images were analyzed using FIJI. For each image, ROIs of layer 4 and layer 2/3 of the visual cortex were defined. The mean gray values were then taken for each ROI. For each mouse, the average mean gray value across both hemispheres was analyzed for both layer 4 and layer 2/3. A two-way ANOVA was performed to test for significance.

For cell-type-specific analysis of candidate gene induction (Fig. 5 and Fig. S5), raw images from the confocal microscope were imported into ImageJ, then the average fluorescence values of *Bdnf*, *Crh*, *Dlx6os1*, *Ephb3* and *Efnb3* within known cell types (defined by marker gene expression within the same tissue) were captured. At least two separate brain sections and 6-10 cells were assessed per biological replicate, with each replicate representing averaged data from one mouse. Data were normalized to the

LDR condition for each biological replicate because it allows us to account for batch effects in fluorescence across rounds of experimentation, and because it provided a direct visualization of the fold change difference in the gene's expression at LDR30m versus LDR.

## Acknowledgements
We thank Dr Timothy J. Burbridge (Boston Children's Hospital), Dr Gabrielle Pouchelon (Cold Spring Harbor Laboratory), Dr Leena Al Ibrahim (King Abdulla University of Science and Technology), Dr Marty Yang (University of California, San Francisco), Samantha Tang (Cold Spring Harbor Laboratory), Justin Park (Cold Spring Harbor Laboratory) and other members of the Cheadle lab for critical input and feedback on the manuscript. We also thank the single-cell biology, sequencing technologies and analysis, and microscopy core facilities at Cold Spring Harbor Laboratory, without which the work would not have been possible.

## Competing interests
The authors declare no competing or financial interests.

## Author contributions
Conceptualization: L.C., A.M.X.; Data curation: A.M.X., C.J.K.; Formal analysis: A.M.X., Q.L.; Funding acquisition: L.C.; Investigation: Q.L.; Methodology: A.M.X., C.J.K.; Project administration: L.C.; Resources: L.C.; Supervision: L.C.; Validation: C.J.K.; Visualization: Q.L., C.J.K.; Writing – original draft: L.C.; Writing – review & editing: L.C., Q.L.

## Funding
This work was supported by the National Institutes of Health - National Institute of Mental Health (R00MH120051 and DP2MH132943 to L.C.), by the National Institutes of Health - National Institute of Neurological Disorders and Stroke (R01NS131486 to L.C.), by a Rita Allen Scholar Award from the Rita Allen Foundation, by a McKnight Scholar Award from the McKnight Foundation, by a Klingenstein-Simons Fellowship Award in Neuroscience from Klingenstein Philanthropies and the Simons Foundation, and by a Brain and Behavior Research Foundation NARSAD grant. Dr Cheadle is a Howard Hughes Medical Institute Freeman Hrabowski Scholar. Open Access funding provided by Cold Spring Harbor Laboratory. Deposited in PMC for immediate release.

## Data availability
Raw and processed snRNAseq data have been deposited in GEO under accession number GSE269482. The source code utilized in the snRNAseq analysis is provided on our laboratory's github page at https://github.com/cheadlelab/snRNA_V1.

## Peer review history
The peer review history is available online at https://journals.biologists.com/dev/lookup/doi/10.1242/dev.204244.reviewer-comments.pdf

## Special Issue
This article is part of the Special Issue 'Lifelong Development: the Maintenance, Regeneration and Plasticity of Tissues', edited by Meritxell Huch and Mansi Srivastava. See related articles at https://journals.biologists.com/dev/issue/152/20.

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
