## [Peer Review File · Development (Cambridge, England)]

A single-cell transcriptomic atlas of sensory-dependent gene expression in developing mouse visual cortex

Andre M. Xavier, Qianyu Lin, Chris J. Kang and Lucas Cheadle

DOI: 10.1242/dev.204244

Editor: Debra L. Silver

Review timeline

Original submission:	12 July 2024
Editorial decision:	31 August 2024
First revision received:	16 January 2025
Accepted:	20 February 2025

Original submission

First decision letter

MS ID#: dev.204244

MS TITLE: A single-cell transcriptomic atlas of sensory-dependent gene expression in developing mouse visual cortex

AUTHORS: Lucas Cheadle, Andre M. Xavier, Qianyu Lin and Chris J. Kang

Dear Dr Cheadle,

I have now received all the referees' reports on the above manuscript, and have reached a decision. The referees' comments are appended below, or you can access them online: please go to:

As you will see, the referees express considerable interest in your work, but have some significant criticisms and recommend a revision of your manuscript before we can consider publication. Both reviewers point out the need for some orthogonal validation, including of neuron type-specific genes. They also raise a number of key points that require clarifications, some of which may involve new experiments.

If you are able to revise the manuscript along the lines suggested, I will be happy receive a revised version of the manuscript. Your revised paper will be re-reviewed by one or more of the original referees, and acceptance of your manuscript will depend on your addressing satisfactorily the reviewers' major concerns. Please also note that Development will normally permit only one round of major revision. If it would be helpful, you are welcome to contact us to discuss your revision in greater detail. Please send us a point-by-point response indicating your plans for addressing the referees' comments, and we will look over this and provide further guidance.

Please attend to all of the reviewers' comments and ensure that you clearly highlight all changes made in the revised manuscript. Please avoid using 'Tracked changes' in Word files as these are lost in PDF conversion. I should be grateful if you would also provide a point-by-point response detailing how you have dealt with the points raised by the reviewers in the 'Response to Reviewers' box. If you do not agree with any of their criticisms or suggestions please explain clearly why this is so.

Reviewer 1:

SUMMARY OF THE ADVANCE MADE IN THIS PAPER AND ITS POTENTIAL SIGNIFICANCE TO THE FIELD

This is a valuable and illuminating study evaluating transcriptional changes in visual cortex development, as a response to sensory input. The analyses performed are robust and the findings are novel. While there are many analyses that highlight the complexity of molecular changes, there is a lack of validation of observations in tissues or functional experiments perturbing signaling relationships. Gain and loss of function experiments in specific cell types seem out of scope for this study, but orthogonal validation that these gene expression changes are meaningful for cell changes, would bolster findings.

SUGGESTIONS TO AUTHORS

Validation of the key observations, using other methods, are needed. Specifically, some validation that there is differential gene expression between specific cell types (excitatory vs inhibitory, L2/3 vs L4) and that key observations, like axon guidance differences, are consistent and robust.

Additional methods details and justification for n=3 mice per condition and how samples were processed (pooled vs separately run) would be informative. In general, the methods could be expanded.

Clarifying key genes that support observations and more thorough description of the findings would be useful. For example, in dense data panels, like Fig 3F and Figs 6C/D, more clarity of the results by including key differences in statements would be helpful.

Additional minor comments/questions:

It's hard to see, due to cell overlap, but it looks like there may be distinct subsets of excitatory neurons in LDR 30mins in Fig 1G. Are those differences meaningful? Is that a single outlier/sample or consistent within that condition?

Does the observation "Jund and Junb both exhibited a pattern of induction more consistent with an LRG identity, peaking at LDR2h rather than LDR30m (Fig. 3A)" conflict with the qPCR data presented in Fig 1C?

In figure 4G-J. What do you make from the shift in overlap where more are unique in L2/3 early and L4 later? Does this suggest anything about the temporal changes in each cell type?

It would be helpful to include the differential axon guidance data described on lines 355-357 in Figure 4 since this is a key finding of this study.

In Figure 5A what does the arrow directionality indicate? I'm more familiar with using RNA velocity as a pseudotime for early developmental cell type transitions. For postmitotic neurons with changes in activity genes, what do these arrows signify biologically and how does this support the induced/repressed observations?

Reviewer 2:

SUMMARY OF THE ADVANCE MADE IN THIS PAPER AND ITS POTENTIAL SIGNIFICANCE TO THE FIELD

In the manuscript by Xavier et al., the authors investigated transcriptional changes linked to sensory stimulation in the visual cortex of post-natal developing mice. They performed single nucleus RNA sequencing (snRNA-seq) in different conditions: normal rearing mice, mice maintained in complete darkness, and mice sensory stimulated for 30 minutes, 2 hours, 4 hours and 6 hours. They sequenced 118,529 single nuclei of neuronal and non-neuronal cell types and identified sensory-driven changes in gene expression in different neuronal sub-types, particularly L2/3 and L4 excitatory neurons. The authors also employed RNA velocity analysis to gather clues on the dynamics of genes expression induction and repression in the context of sensory stimulation. Finally, the authors investigated potential cell-cell interactions by studying the expression of known ligand-receptor pairs between the different neuronal types. The data shown in this manuscript represent valuable resources that may be used to further advance the field and our understanding of sensory stimuli dependent molecular changes. Nevertheless, I have several major and minor concerns that should be addressed prior to publication in Development.

SUGGESTIONS TO AUTHORS

Major comments:

1. The authors state that excitatory neurons, particularly L2/3 and L4 neurons, show the most differences in gene expression upon sensory stimulation. However, based on figure 1-F it seems that more nuclei of the L2/3 and L4 subtypes were sequenced compared to the other cell types. Could the greater number of L2/3 and L4 neurons sequenced bias the differential gene expression analysis? Were the data normalized prior to the analysis (i.e., down-sampling reads from different cell types)? Also, I did not receive the tables mentioned in the text and therefore did not have access to table 1 describing the numbers of cells analyzed, or table 2 describing the numbers of differentially expressed genes across cell types.
2. In Figure 3B-E, the authors use in situ hybridization to claim that L2/3 and L4 neurons show the strongest induction of Fos and Nr4a1 compared to other cell types. However, the other cell types are not labeled or quantified. The authors should show the entire brain section and not only the part of the image showing L2/3 and L4, and should include quantification of Fos and Nr4a1 induction in other cell types to strengthen their conclusion.
3. While the RNA velocity analysis in figure 5 is informative and interesting, strong conclusions should not be drawn out of these results. The authors sequenced nuclear RNA, so it is unlikely that much post-transcriptional regulation has occurred. In addition, the library preparation method used by the authors enriches for polyadenylated RNA, thus the information on nascent RNA (introns) is incomplete. As a result, the comparison of RNA velocity results to DGE analysis is inadequate for making conclusions regarding transcription-independent regulation. The authors could instead utilize the degradation rate of mRNA from the RNA velocity analysis to interrogate potential post-transcriptional regulation, but again, the amount of post-transcriptional regulation captured in this dataset is likely limited because nuclear RNA was sequenced. These limitations should be discussed in the manuscript.
4. For the RNA velocity analysis in Figure 5A, it would be more informative if the authors showed a PCA with all the different timepoints together, rather than multiple pair-wise comparisons. This would better represent the progression between the different states of stimulation. They should include LDR, all LDR stimulation timepoints, and NR in this analysis.
5. The value of this resource would be strengthened if the authors validated some of the new gene expression changes by in situ hybridization. The authors performed qPCR and in situ hybridization to confirm the expected increase in Fos and Jun expression, but it would greatly benefit the manuscript if the authors validated other less known genes, especially those that show neuronal type-specific changes in expression at the different timepoints after stimulation.
6. Following the previous point, it would also be interesting to validate the proposed cell-cell interactions identified in figures 6 and 7. This can be done by immunostaining brain sections, for example, and looking at the expression of ligands and receptors in cells that are in proximity.
7. The authors sequenced 118,529 single nuclei of neuronal and non-neuronal cell types yet mostly focused on L2/3 and L4 neurons in their analysis. I suggest mentioning their findings on other cell types as well, particularly non-neuronal cells, as this can increase the impact of their manuscript and the value of their snRNA-seq. For instance, the number of upregulated and downregulated genes in the other neuronal and non-neuronal cell types could be added to Figure 2 as well as Figure 3G.
8. The authors mention several times that single-cell analysis of sensory-dependent stimulation has been previously performed in adult mice. How do the changes in genes expression after sensory stimulation at this developmental stage compare with published data in adult mice? It would be interesting to include such a comparison in the manuscript and highlight potential similarities and/or differences between the developing and the adult mouse neurons' response to stimulation.

Minor comments:

1. The tables mentioned in the text were not included in the file for reviewers.
2. Line 28: The term "unique" used to define the sensory-induced genes can be misinterpreted by the reader and should be explained better.
3. Line 111: The word cohort in "four cohorts of sensory stimulated mice" implies biological replicates rather than timepoints. Maybe replace with a more suitable word, like "timepoints".
4. Paragraph from line 200 to 226: The way the authors describe the changes in expression in this paragraph is confusing and difficult to follow. The authors should consider describing changes in a relative manner to make it easier for the reader. For instance, replace "less highly expressed" and "more highly expressed" by downregulated and upregulated respectively.
5. Page 6 is missing references, for example: lines 237-238 and lines 261-262.
6. In figure 3H-I, the authors suggest that there is more overlap of late genes than early genes between excitatory and inhibitory neurons, in contrast to previous studies in adult mice (lines 261-262). This conclusion would be strengthened if the authors compared the early and late genes between more cell types.
7. The venn diagrams in figures 3H, 4G-J, and 5D-K should include statistics. In addition, figure 5D-K would be easier to follow if the comparisons made in the venn diagrams were labeled.
8. In figure 2, the authors compared the neuronal transcriptomes of LDR and NR and found relatively few numbers of differentially expressed genes and surprisingly most of these genes were upregulated in LDR compared to NR. However, when the authors compared LDR to LDR30min and LDR4h, most genes were found to be downregulated in LDR. Based on these results, it would be interesting to compare the gene expression changes between the different timepoints after stimulation to NR.
9. The CellChat analyses performed in figures 6 and 7 are in the normal rearing condition. It would be interesting to analyze whether these cell-cell interactions change with stimulation.

First revisionAuthor response to reviewers' comments

Response to reviewers

We thank the reviewers for their helpful feedback which has allowed us to improve the rigor and scope of the study. We discuss how we have addressed their specific comments below.

Reviewer 1: SUMMARY OF THE ADVANCE MADE IN THIS PAPER AND ITS POTENTIAL SIGNIFICANCE TO THE FIELD

This is a valuable and illuminating study evaluating transcriptional changes in visual cortex development, as a response to sensory input. The analyses performed are robust and the findings are novel. While there are many analyses that highlight the complexity of molecular changes, there is a lack of validation of observations in tissues or functional experiments perturbing signaling relationships. Gain and loss of function experiments in specific cell types seem out of scope for this study, but orthogonal validation that these gene expression changes are meaningful for cell changes, would bolster findings.

We thank the reviewer for their assessment of the study and are pleased that they find it to be illuminating, robust, and novel. We agree that additional orthogonal validations and insights into

the functional relevance of the gene programs we identify would strengthen the manuscript, and we describe how we have now addressed this comment in the revised submission below.

SUGGESTIONS TO AUTHORS

Validation of the key observations, using other methods, are needed. Specifically, some validation that there is differential gene expression between specific cell types (excitatory vs inhibitory, L2/3 vs L4) and that key observations, like axon guidance differences, are consistent and robust.

We agree that orthogonally validating the cell-type-specific induction of some of the sensory-dependent genes identified in our dataset would strengthen the rigor and more fully support the conclusions of the study. We have now employed multiplexed fluorescence *in situ* hybridization (FISH) using the RNAscope platform to validate the induction of the following genes, each of which is induced in some cell types but not others:

Brain-derived neurotrophic factor (BDNF): we have validated that the well-known synaptic modulator *BDNF* is induced by experience in layer 4 (L4) excitatory neurons but not in VIP+ inhibitory neurons.

Corticotropin-releasing hormone (Crh): we have validated that the stress-associated molecule *Crh* is induced by experience in VIP+ inhibitory neurons but not in L4 excitatory neurons, NPY+ inhibitory neurons, or PV+ inhibitory neurons.

Dlx6os1: we have validated that the long non-coding RNA *Dlx6os1* is induced by experience in VIP+ and NPY+ inhibitory neurons but not in L4 excitatory or PV+ inhibitory neurons.

These new data make up revised Figure 5 and are described in lines 346-355 of the text.

In addition, based upon the reviewer's interest in the axon guidance findings, we have also validated the induction of the axon guidance receptor *Ephb3* in L2/3 but not L4 neurons. We have also mapped expression of the EphB3 ligand *Efnb3* across the cortical layers, finding that it is more highly expressed in layers 4-6 than in layers 2/3, consistent with L2/3 neurons utilizing this Eph/Ephrin signaling pathway to make axonal projections onto neurons within deeper layers of V1. These results are now included as Supplemental Figure 5 and discussed in lines 318-328 of the text.

Additional methods details and justification for n=3 mice per condition and how samples were processed (pooled vs separately run) would be informative. In general, the methods could be expanded.

We have now expanded the methods to provide more experimental detail as suggested including a discussion of the choice to perform three biological replicates. In brief, the reasons for this are two-fold. First, it is the standard in the field that single-cell RNA-sequencing experiments should include at least 3 distinct biological replicates to control for batch effects, among other reasons. The second reason for including 3 replicates is based upon my prior experience utilizing these approaches, I have found that 3 replicates can typically provide sufficient data for supporting robust and consistent conclusions that can be validated using orthogonal approaches. Updated descriptions in the methods section can be found in the following lines: 583-588; 604-611; 644-647; and 720-729.

Clarifying key genes that support observations and more thorough description of the findings would be useful. For example, in dense data panels, like Fig 3F and Figs 6C/D, more clarity of the results by including key differences in statements would be helpful.

We have now expanded the text around these results in response to the suggestion (especially Fig. 3F which is now Fig. 2F), and we include a new main figure describing transcriptomic changes in inhibitory neurons and glial cells to complement the discussion of changes in excitatory neurons. See lines 243-257 and 330-379 of the revised text alongside new figure 4.

Additional minor comments/questions:

It's hard to see, due to cell overlap, but it looks like there may be distinct subsets of excitatory neurons in LDR 30mins in Fig 1G. Are those differences meaningful? Is that a single outlier/sample or consistent within that condition?

In the UMAP in Figure 1G, we agree that it appears that there may be a subset of L2/3 excitatory neurons that derive largely from cells within the LDR30 condition, which would be consistent with our finding that L2/3 excitatory neurons are particularly susceptible to sensory experience at the transcriptional level. To further explore this possibility, we subset the 27,482 L2/3 neurons in the dataset and performed an additional round of clustering on this population alone. This experiment identified six sub-clusters of L2/3 neurons, 3 of which (clusters 0, 1, and 3) included relatively similar numbers of cells across conditions. The other 3 clusters were each predominantly made up by cells from 1 - 2 conditions as follows: cluster 2 is principally composed of cells from the LDR2 and LDR4 conditions (87%); cluster 4 is primarily made up of cells from the LDR30 condition (78%); and cluster 5 is primarily made up of cells from the LDR6 condition (97%).

We next asked whether these observations were consistent across the three biological replicates by quantifying the number of cells from each replicate that falls into each of these clusters. For cluster 2, 1028, 1604, and 1475 cells were derived from bioreps 1, 2, and 3, respectively, suggesting that the results reflect consistency across replicates. For cluster 4 which is enriched for LDR30m cells, 877, 821, and 536 cells derived from bioreps 1, 2, and 3 respectively, again suggesting consistency across replicates. In contrast, cluster 5 is largely made up of cells from biological replicate 1 and is therefore less likely to represent a meaningful distinction as it is not consistent across replicates. Thus, we conclude that visual stimulation drives changes in gene expression in some L2/3 neurons that are extreme enough to give rise to distinct transcriptomic states. We now include these data in Supplemental Figure 4 and discuss the results in the revised text, lines 226-239 (UMAPs included below for your convenience).

NOTE: We have removed unpublished data that had been provided for the referees in confidence.

Does the observation "Junb and Junb both exhibited a pattern of induction more consistent with an LRG identity, peaking at LDR2h rather than LDR30m (Fig. 3A)" conflict with the qPCR data presented in Fig 1C?

The data do not conflict because, while all of these genes belong to the same family of AP1 transcription factors, *Jun* as analyzed in Figure 1 is a different gene than *Junb* and *Junb*. We now clarify this in the text, lines 214-217.

In figure 4G-J. What do you make from the shift in overlap where more are unique in L2/3 early and L4 later? Does this suggest anything about the temporal changes in each cell type?

This could certainly reflect a difference in the temporal aspects of sensory-driven transcription in these cells, although I don't think the differences are big enough to draw strong conclusions based upon this analysis.

It would be helpful to include the differential axon guidance data described on lines 355-357 in Figure 4 since this is a key finding of this study.

We agree with the reviewer that the axon guidance finding is one of the more interesting findings in the study. We have now further investigated this as described in our response to the reviewer's point 1 above. In brief, we have added a supplemental figure to the manuscript validating *Ephb3* induction in L2/3 but not L4 neurons following stimulation, and showing that the ligand of *Ephb3*, *Efnb3*, is most highly expressed in deeper layers of cortex known to receive inputs from L2/3 neurons. See Supplemental Figure 5 and lines 318-328 in the text.

In Figure 5A what does the arrow directionality indicate? I'm more familiar with using RNA velocity as a pseudotime for early developmental cell type transitions. For postmitotic neurons with changes in activity genes, what do these arrows signify biologically and how does this support the induced/repressed observations?

The vectors in the RNA velocity figure (original Figure 5A, now Figure 6A) represent inferred future transcriptional states of neurons based on spliced and unspliced RNA dynamics between any two given conditions. These arrows indicate the direction and magnitude of transcriptional changes, in our case, reflecting how neurons respond to light stimulation. Specifically, the vectors highlight transcriptional shifts toward activity-induced transcriptomic states during stimulation and away from repressed transcriptional states post-stimulation. This allows us to visualize the dynamic transitions in gene expression triggered by external stimuli.

RNA velocity differs fundamentally from pseudotime analysis. While pseudotime orders cells along a developmental trajectory as the reviewer indicates, emphasizing lineage progression, RNA velocity provides a snapshot of real-time transcriptional dynamics by leveraging the temporal relationships between spliced and unspliced RNA. Pseudotime assumes a continuous trajectory, which is well-suited for developmental studies but less applicable to contexts like ours, where the focus is on more acute, stimulus-driven changes rather than developmental progression.

As you can see below, Reviewer #2 brought up several important caveats for the RNA velocity analysis, and we have modified our discussion of these results in the revised text accordingly. Please see lines 391-395 and 525-537 for additional clarifications and a discussion of the caveats involved in interpreting these findings. Please also keep in mind that the RNA velocity figure has moved from Figure 5 to Figure 6.

Reviewer 2: SUMMARY OF THE ADVANCE MADE IN THIS PAPER AND ITS POTENTIAL SIGNIFICANCE TO THE FIELD

In the manuscript by Xavier et al., the authors investigated transcriptional changes linked to sensory stimulation in the visual cortex of post-natal developing mice. They performed single nucleus RNA sequencing (snRNA-seq) in different conditions: normal rearing mice, mice maintained in complete darkness, and mice sensory stimulated for 30 minutes, 2 hours, 4 hours and 6 hours. They sequenced 118,529 single nuclei of neuronal and non-neuronal cell types and identified sensory-driven changes in gene expression in different neuronal sub-types, particularly L2/3 and L4 excitatory neurons. The authors also employed RNA velocity analysis to gather clues on the dynamics of genes expression induction and repression in the context of sensory stimulation. Finally, the authors investigated potential cell-cell interactions by studying the expression of known ligand-receptor pairs between the different neuronal types. The data shown in this manuscript represent valuable resources that may be used to further advance the field and our understanding of sensory stimuli dependent molecular changes. Nevertheless, I have several major and minor concerns that should be addressed prior to publication in Development.

We thank the reviewer for their helpful comments and suggestions, which we address one-by-one below.

SUGGESTIONS TO AUTHORS

Major comments:

1. The authors state that excitatory neurons, particularly L2/3 and L4 neurons, show the most differences in gene expression upon sensory stimulation. However, based on figure 1-F it seems that more nuclei of the L2/3 and L4 subtypes were sequenced compared to the other cell types. Could the greater number of L2/3 and L4 neurons sequenced bias the differential gene expression analysis? Were the data normalized prior to the analysis (i.e., down-sampling reads from different cell types)? Also, I did not receive the tables mentioned in the text and therefore did not have access to table 1 describing the numbers of cells analyzed, or table 2 describing the numbers of differentially expressed genes across cell types.

We thank the reviewer for raising this important consideration. First, we did provide the tables upon submission and are sorry to hear that they did not make their way to the reviewers. We believe these are an important part of the paper as they provide key information without requiring a complete re-analysis of the data, and we made a note in the new cover letter to please provide the reviewers with access to them along with the revised manuscript.

To address the reviewer's question, in the initial analysis, we did not perform any downsampling to normalize cell numbers prior to identifying differentially expressed genes (DEGs). We agree, however, that the number of nuclei for a given cell type can have the potential to influence the number of DEGs identified. Thus, in the revised manuscript, we have added two supplemental figures addressing this comment. The first figure, Supplemental Figure 2, provides a clearer view of the numbers of genes upregulated or downregulated across cell types in our dataset prior to downsampling or normalization across cell types. In the second new figure, Supplemental Figure 3, we compared the numbers of DEGs for the most abundant cell types in the dataset without downsampling (panel A) and after downsampling each cell type to 400 cells per condition (panel B). This experiment validated that L2/3 neurons display the highest number of DEGs among cell types analyzed both before and after downsampling, which is a key finding in our study. The results also highlight that, in addition to L4 neurons, L6 excitatory populations exhibit high numbers of DEGs both before and downsampling. In addition to new supplemental figures 2 and 3, please see lines 217-224 in the revised text.

2. In Figure 3B-E, the authors use *in situ* hybridization to claim that L2/3 and L4 neurons show the strongest induction of *Fos* and *Nr4a1* compared to other cell types. However, the other cell types are not labeled or quantified. The authors should show the entire brain section and not only the part of the image showing L2/3 and L4, and should include quantification of *Fos* and *Nr4a1* induction in other cell types to strengthen their conclusion.

We appreciate this comment as it has allowed us to clarify the rationale around this experiment and result. The *Fos* and *Nr4a1* FISH analysis which can now be found in Figure 2B-E is meant as a broad validation that these genes are highly induced across numerous layers and cell types in visual cortex following stimulation. In other words, it is meant as a simple validation that light re-exposure indeed elicits robust expression of canonical IEGs *Fos* and *Jun* across cortical layers which can be observed not only through sequencing but in tissue as well. Please see revised manuscript lines 211-214.

We also recognize the importance of orthogonal validations of DEG induction *in situ* and have added such validations to new Figure 5 and Supplemental Figure 5 as described in our response to reviewer #1 above, and as detailed below as well.

3. While the RNA velocity analysis in figure 5 is informative and interesting, strong conclusions should not be drawn out of these results. The authors sequenced nuclear RNA, so it is unlikely that much post-transcriptional regulation has occurred. In addition, the library preparation method used by the authors enriches for polyadenylated RNA, thus the information on nascent RNA (introns) is incomplete. As a result, the comparison of RNA velocity results to DGE analysis is inadequate for making conclusions regarding transcription-independent regulation. The authors could instead utilize the degradation rate of mRNA from the RNA velocity analysis to interrogate potential post-transcriptional regulation, but again, the amount of post-transcriptional regulation captured in this dataset is likely limited because nuclear RNA was sequenced. These limitations should be discussed in the manuscript.

We thank the reviewer for their insightful comments on the RNA velocity analysis presented in Figure 5 (now Figure 6) and its limitations due to the use of nuclear RNA-sequencing in this study. We acknowledge that nuclear RNA-sequencing primarily captures pre-mRNA, limiting our ability to study post-transcriptional regulation, as much of this occurs in the cytoplasm. Furthermore, we agree that the enrichment for polyadenylated RNA in our library preparation method reduces the representation of nascent RNAs (i.e. intronic reads), which are critical for robust RNA velocity modeling.

We have now provided greater discussion of this analysis and its caveats in lines 391-395 and 525-537 of the revised text, and we have softened our conclusions around the RNA velocity results. We do note that other studies have been able to harness snRNAseq data to derive insights from RNA velocity analyses, and we cite those papers as well in references 46 and 47.

4. For the RNA velocity analysis in Figure 5A, it would be more informative if the authors showed a PCA with all the different timepoints together, rather than multiple pair-wise comparisons. This

would better represent the progression between the different states of stimulation. They should include LDR, all LDR stimulation timepoints, and NR in this analysis.

Your suggestion aligns with the conceptual aim of the RNA velocity analysis. In initially performing this analysis, we indeed included all conditions from NR to LDR6h together. However, when analyzing all timepoints together, the resulting trajectories failed to follow the expected sequence. This could be due to overlapping signals or complexities in modeling transitions across multiple conditions simultaneously, especially in a case like this where we are studying acute changes in transcription between pairs of timepoints which are likely to be less extreme than those observed in the context of developmental lineage progression. In contrast, pairwise comparisons effectively resolve the dynamics between adjacent timepoints, accurately reflecting the expected progression from one stimulation timepoint to the other. This is why we opted for pairwise analysis rather than combining all timepoints.

5. The value of this resource would be strengthened if the authors validated some of the new gene expression changes by in situ hybridization. The authors performed qPCR and in situ hybridization to confirm the expected increase in Fos and Jun expression, but it would greatly benefit the manuscript if the authors validated other less known genes, especially those that show neuronal type-specific changes in expression at the different timepoints after stimulation.

We completely agree with the reviewer about the value of orthogonal validations and have added new experimental data supporting the cell-type-specific induction of 4 DEGs to the manuscript in new Figure 5 and Supplemental Figure 5. As described above in response to Reviewer #1, these changes are the following:

Brain-derived neurotrophic factor (BDNF): we have validated that the well-known synaptic modulator *BDNF* is induced by experience in layer 4 (L4) excitatory neurons but not in VIP+ inhibitory neurons.

Corticotropin-releasing hormone (Crh): we have validated that the stress-associated molecule *Crh* is induced by experience in VIP+ inhibitory neurons but not in L4 excitatory neurons, NPY+ inhibitory neurons, or PV+ inhibitory neurons.

Dlx6os1: we have validated that the long non-coding RNA *Dlx6os1* is induced by experience in VIP+ and NPY+ inhibitory neurons but not in L4 excitatory or PV+ inhibitory neurons.

These new data make up revised Figure 5 and are described in lines 346-355 of the text.

In addition, we have also validated the induction of the axon guidance receptor *Ephb3* in L2/3 but not L4 neurons. We have also mapped expression of the EphB3 ligand *Efnb3* across the cortical layers, finding that it is more highly expressed in layers 4-6 than in layers 2/3, consistent with L2/3 neurons utilizing this Eph/Ephrin signaling pathway to make axonal projections onto neurons within deeper layers of V1. These results are now included as Supplemental Figure 5 and discussed in lines 318-328 of the text.

6. Following the previous point, it would also be interesting to validate the proposed cell-cell interactions identified in figures 6 and 7. This can be done by immunostaining brain sections, for example, and looking at the expression of ligands and receptors in cells that are in proximity.

We appreciate this suggestion to bolster the results of the *CellChat* analysis. We attempted to optimize antibody staining approaches to capture neuroligin and neurexin protein expression in an effort to correlate this with our RNA-sequencing results, given that these pathways are implicated by our *CellChat* analysis. Unfortunately, immunostaining for these proteins has apparently been a major challenge in the field and we were unable to get the stainings to work in our hands. Thus, we instead focused on adding additional insights to the axon guidance results as described above. We hope the reviewer will agree that these experiments have added more functional insights to the manuscript, albeit without an explicit focus on the *CellChat* results.

7. The authors sequenced 118,529 single nuclei of neuronal and non-neuronal cell types yet mostly focused on L2/3 and L4 neurons in their analysis. I suggest mentioning their findings on other cell types as well, particularly non-neuronal cells, as this can increase the impact of their manuscript and the value of their snRNA-seq. For instance, the number of upregulated and downregulated genes in the other neuronal and non-neuronal cell types could be added to Figure 2 as well as Figure 3G.

We agree that a more extensive discussion of genes induced in other neuronal and non-neuronal cell types beyond L2/3 and L4 excitatory neurons would enrich the study. We have now added data describing the responses of the 4 subclasses of inhibitory neurons as well as the three best-represented populations of glia (astrocytes, oligodendrocytes, and microglia) to sensory experience. We have added two relevant figures. First, we added Supplemental Figure 2 which includes the total numbers of genes that were upregulated (Fig. S2A) or downregulated (Fig. S2B) by experience across excitatory, inhibitory, and glial cell types in the dataset. Second, we added a new main Figure 4 which provides volcano plots of sensory-induced transcription across inhibitory and glial cells. Also note that new Figure 5 includes several validations of gene expression in VIP+, NPY+, and PV+ interneurons. We hope the reviewer will agree that these additions provide a more comprehensive view of the data. In addition to Figures S2, 4, and 5, see lines 243-248 and 330-379.

8. The authors mention several times that single-cell analysis of sensory-dependent stimulation has been previously performed in adult mice. How do the changes in genes expression after sensory stimulation at this developmental stage compare with published data in adult mice? It would be interesting to include such a comparison in the manuscript and highlight potential similarities and/or differences between the developing and the adult mouse neurons' response to stimulation.

This is a great suggestion although it is a little tricky to address. The most relevant dataset for comparison with ours is found in Hrvatin*, Hochbaum*, and Nagy* et al, *Nature Neuroscience*, 2018 (full disclosure: I am a co-author on this paper). Here, the authors performed single-cell RNA-sequencing on the breadth of cells within mouse visual cortex following late-dark-rearing (LDR) and visual stimulation in adult mice. They employed the exact same LDR paradigm as that used in our study, however they looked at fewer timepoints, focusing only on LDR0, LDR1h, and LDR4h. In my view, two inconsistencies between the studies that may hamper our ability to draw strong conclusions from a comparison are (1) the Hrvatin study employed whole-cell sequencing whereas we utilized nuclear sequencing, and (2) the Hrvatin study chose 1 hour post-stimulation as an early response time point while we performed this analysis at 30 minutes post-stimulation. While we feel there is value in comparing the datasets regardless of these differences, integrating the datasets and applying a new range of computational analyses to it is likely beyond the scope of the current study. See new discussion of this point in lines 539-549.

Minor comments:

1. The tables mentioned in the text were not included in the file for reviewers.

It is unfortunate that the tables we uploaded along with the submission were not provided to the reviewers. We have explicitly asked for the tables to be shared in the cover letter attached to this revised manuscript.

2. Line 28: The term "unique" used to define the sensory-induced genes can be misinterpreted by the reader and should be explained better.

We feel the word "unique" is unnecessary here so we have removed it from the manuscript.

3. Line 111: The word cohort in "four cohorts of sensory stimulated mice" implies biological replicates rather than timepoints. Maybe replace with a more suitable word, like "timepoints".

For clarity, we have removed the statement from the revised paper.

4. Paragraph from line 200 to 226: The way the authors describe the changes in expression in this paragraph is confusing and difficult to follow. The authors should consider describing changes in a

relative manner to make it easier for the reader. For instance, replace "less highly expressed" and "more highly expressed" by downregulated and upregulated respectively.

We have now made these changes, see lines 172-201.

5. Page 6 is missing references, for example: lines 237-238 and lines 261-262.

Reference for lines 237-238 (see new lines 211-214) added. Lines 261-262 were removed from the text.

6. In figure 3H-I, the authors suggest that there is more overlap of late genes than early genes between excitatory and inhibitory neurons, in contrast to previous studies in adult mice (lines 261-262). This conclusion would be strengthened if the authors compared the early and late genes between more cell types.

We agree this would be informative. However, due to space limitations, we chose to focus the revision on what we viewed as more pressing issues (e.g. orthogonal validations of induced genes). We hope the reviewer will agree that the nature and relevance of overlap in gene programs among cell populations could be a nice starting place for a follow-up study.

7. The venn diagrams in Figures 3H, 4G-J, and 5D-K should include statistics. In addition, figure 5D-K would be easier to follow if the comparisons made in the venn diagrams were labeled.

Thank you for your feedback regarding the Venn diagrams in original Figures 3H, 4G-J, and 5D-K. We'd like to clarify that these Venn diagrams are not meant as statistical comparisons but are rather meant to provide a visual representation of overlapping gene programs between cohorts. In original Figure 5D-K (new Figure 6D-I) we labeled the circles for panels D and H which we hope will help with clarity.

8. In figure 2, the authors compared the neuronal transcriptomes of LDR and NR and found relatively few numbers of differentially expressed genes and surprisingly most of these genes were upregulated in LDR compared to NR. However, when the authors compared LDR to LDR30min and LDR4h, most genes were found to be downregulated in LDR. Based on these results, it would be interesting to compare the gene expression changes between the different timepoints after stimulation to NR.

In the revised manuscript, we include a new Table, Table 3, which lists genes that were found to be differentially expressed between each stimulation timepoint and the NR condition. This complements Table 2 which includes comparisons between stimulation timepoints and the LDR conditions. Also see the revised text, lines 198-201.

9. The CellChat analyses performed in figures 6 and 7 are in the normal rearing condition. It would be interesting to analyze whether these cell-cell interactions change with stimulation.

We initially performed *CellChat* at each timepoint as the reviewer suggests; however, we did not observe strong differences between stimulation timepoints. We believe this is likely to be a feature of the analysis being informative but not particularly precise. Thus, we focused the analysis in the paper on the normally reared (NR) timepoint to provide insights into how cells in V1 may be interacting under normal physiological conditions during a critical period of sensory-dependent circuit development.

Second decision letter

MS ID#: dev.204244R1

MS TITLE: A single-cell transcriptomic atlas of sensory-dependent gene expression in developing mouse visual cortex

AUTHORS: Lucas Cheadle, Andre M. Xavier, Qianyu Lin and Chris J. Kang

Dear Dr Cheadle,

I am happy to tell you that your manuscript has been accepted for publication in Development, pending our standard publication integrity checks.